# Structural basis for directional chitin biosynthesis

Wei Chen[1,2,8], Peng Cao[3,8], Yuansheng Liu[4,8], Ailing Yu[2], Dong Wang[4], Lei Chen[2], Rajamanikandan Sundarraj[5], Zhiguang Yuchi[5], Yong Gong[6✉], Hans Merzendorfer[7] & Qing Yang[1,2,4✉]

Chitin, the most abundant aminopolysaccharide in nature, is an extracellular polymer consisting of *N*-acetylglucosamine (GlcNAc) units[1]. The key reactions of chitin biosynthesis are catalysed by chitin synthase[2–4], a membrane-integrated glycosyltransferase that transfers GlcNAc from UDP-GlcNAc to a growing chitin chain. However, the precise mechanism of this process has yet to be elucidated. Here we report five cryo-electron microscopy structures of a chitin synthase from the devastating soybean root rot pathogenic oomycete *Phytophthora sojae* (*Ps*Chs1). They represent the apo, GlcNAc-bound, nascent chitin oligomer-bound, UDP-bound (post-synthesis) and chitin synthase inhibitor nikkomycin Z-bound states of the enzyme, providing detailed views into the multiple steps of chitin biosynthesis and its competitive inhibition. The structures reveal the chitin synthesis reaction chamber that has the substrate-binding site, the catalytic centre and the entrance to the polymer-translocating channel that allows the product polymer to be discharged. This arrangement reflects consecutive key events in chitin biosynthesis from UDP-GlcNAc binding and polymer elongation to the release of the product. We identified a swinging loop within the chitin-translocating channel, which acts as a 'gate lock' that prevents the substrate from leaving while directing the product polymer into the translocating channel for discharge to the extracellular side of the cell membrane. This work reveals the directional multistep mechanism of chitin biosynthesis and provides a structural basis for inhibition of chitin synthesis.

The biosynthesis of chitin is essential for the survival and reproduction of various organisms from different taxonomic groups, such as life-threatening fungi, agriculture-devastating oomycetes and insect pests. Therefore, it provides a preferred target for discovering antifungal agents or pesticides[4–6].

The core of the chitin biosynthetic machinery is an integral membrane enzyme named chitin synthase (CHS) (EC 2.4.1.16)[3]. CHS belongs to glycosyltransferase family 2 (GT2), a large enzyme family that includes cellulose, alginate and hyaluronan synthases[7,8]. Chitin synthesis is proposed to involve three major steps: (1) the processive addition of GlcNAc from UDP-GlcNAc (donor substrate) to the terminal C4-hydroxyl group of the nascent chitin chain (acceptor substrate) by the catalytic domain of the enzyme facing the cytoplasmic side; (2) the release of the nascent chain to the extracellular space through a transmembrane channel within the enzyme; and (3) the spontaneous assembly of released nascent chains into nanofibrils[3]. CHS controls the first two steps of this process but may also participate in the formation of fibrils. Despite differing in the number of transmembrane helices and organization of the respective cytosolic domains, CHSs from various species share a conserved catalytic domain[9,10] (Extended Data Fig. 1), thus allowing the development of competitive inhibitors with broad-spectrum activities. Because chitin is absent in plants and mammals, CHS might constitute one of the safest among the 30 currently used insecticidal and fungicidal targets for the control of fungal pathogens and insect pests[11]. Among the fungicidal agents that target CHS is nikkomycin Z (NikZ), which consists of a pyrimidine-nucleoside peptide backbone and is a first-generation broad-spectrum CHS inhibitor currently in phase II clinical trials[12,13].

*P. sojae* is a pathogen that causes soybean (*Glycine max* L.) root and stem rot, which results in economic losses of more than US$1 billion per year[14,15]. Knockout of the *P. sojae* chitin synthase *Ps*Chs1 impairs mycelial growth, sporangial production and zoospore release, and thus greatly reduces the virulence of *P. sojae*[16]. *Ps*Chs1 serves as both an excellent antifungal target and a model system for CHS research. In this study, we report five cryo-electron microscopy (cryo-EM) structural snapshots of *Ps*Chs1, which provide not only a mechanistic understanding of chitin biosynthesis at the atomic level but also a structural basis for the rational design of CHS-targeting inhibitors.

[1]State Key Laboratory for Biology of Plant Diseases and Insect Pests, Institute of Plant Protection, Chinese Academy of Agricultural Sciences, Beijing, China. [2]Shenzhen Branch, Guangdong Laboratory of Lingnan Modern Agriculture, Genome Analysis Laboratory of the Ministry of Agriculture and Rural Affairs, Agricultural Genomics Institute at Shenzhen, Chinese Academy of Agricultural Sciences, Shenzhen, China. [3]Faculty of Environment and Life, Beijing University of Technology, Beijing, China. [4]School of Bioengineering, Dalian University of Technology, Dalian, China. [5]Tianjin Key Laboratory for Modern Drug Delivery and High-Efficiency, Collaborative Innovation Center of Chemical Science and Engineering, School of Pharmaceutical Science and Technology, Tianjin University, Tianjin, China. [6]Center for Multi-disciplinary Research, Institute of High Energy Physics, Chinese Academy of Sciences, Beijing, China. [7]Department of Chemistry and Biology, School of Science and Technology, University of Siegen, Siegen, Germany. [8]These authors contributed equally: Wei Chen, Peng Cao, Yuansheng Liu. ✉e-mail: yonggong@ihep.ac.cn; qingyang@caas.cn

## Enzymatic activity

*Ps*Chs1 was mainly purified as a dimer in solution (Extended Data Fig. 2a–c). Activity of *Ps*Chs1 clearly depends on specific divalent ions, and EDTA completely blocked enzyme activity (Extended Data Fig. 2d). The addition of GlcNAc together with divalent ions significantly increased the activity of *Ps*Chs1 (Extended Data Fig. 2d). Enzyme kinetics revealed a Hill coefficient of 1, indicating that GlcNAc is not a positive effector of *Ps*Chs1 (Extended Data Fig. 2e). In line with previous data, which has shown that yeast chitin synthase Chs2 can use 2-acylamido analogues of GlcNAc as acceptors of GlcNAc derived from UDP-GlcNAc[17], this finding suggests that free GlcNAc may act as an acceptor to prime the reaction. Of note, the addition of (GlcNAc)$_{2-5}$ did not affect enzyme activity (Extended Data Fig. 2f).

The sugar polymer produced by *Ps*Chs1 could be degraded by *Ostrinia furnacalis* Chi-h, a chitinase that specifically hydrolyses chitin, confirming that the product is chitin (Extended Data Fig. 2d). Using a scanning electron microscope, we observed that the synthesized chitin appeared as a fibrous material, and the amount of chitin fibre increased as the reaction time progressed (Extended Data Fig. 2g). Under a confocal laser scanning microscope, chitin was specifically detected by wheat germ agglutinin coupled to the fluorophore fluorescein isothiocyanate. It appeared as aggregated fibrillar material at high magnification by a scanning electron microscope, but as a 'roundish' soft material at lower magnification by a confocal laser scanning microscope (Extended Data Fig. 2h). The isomorphic type of the synthesized chitin was determined by X-ray diffraction and attenuated total reflectance Fourier transform infrared (ATR-FTIR) spectroscopy (Extended Data Fig. 2i). As indicated in ATR-FTIR spectroscopy, the synthesized chitin showed the same adsorption spectrum as shrimp α-chitin, where the characteristic C=O stretching (amide I) band at 1,620–1,670 cm$^{-1}$ appeared as a doublet, which was clearly distinguishable from a singlet that appeared in the crystalline β-chitin[18–20]. The ATR-FTIR spectra are in line with the X-ray diffraction results. The four sharp diffraction peaks of synthesized chitin observed at 9.3°, 12.7°, 19.3° and 26.4°, which corresponded to the 020, 021, 110 and 013 planes, respectively, are typical crystal patterns of α-chitin[21,22]. Therefore, the polymer synthesized by *Ps*Chs1 is α-chitin, which is consistent with data in the literature that have demonstrated that chitin formed in oomycete species is of the α-type[23].

## Architecture of *Ps*Chs1

The different cryo-EM structures of *Ps*Chs1 were reconstructed by imposing a C2 symmetry and reached overall resolutions of 3.1 Å (UDP bound), 3.2 Å (NikZ bound), 3.3 Å (apo and UDP-GlcNAc bound) and 3.9 Å (UDP/(GlcNAc)$_3$ bound) (Extended Data Table 1 and Extended Data Figs. 3–7). The EM maps were of sufficient quality to allow de novo building of residues 40–860, with a disordered region of residues 743–758, in all five structures. The donor substrate-bound structure shows an additional N-terminal region from residues 23–39 (Extended Data Fig. 4e). All the structures include the N-terminal domain (NTD), the glycosyltransferase (GT) domain and all α-helices of the C-terminal transmembrane (TM) domain (Fig. 1a–c). The TM region comprises a cluster of six TM helices (TM1–6) that reside on top of three amphipathic interface helices (IF1–3) located at the boundary between the membrane and the cytosol (Fig. 1c and Extended Data Fig. 8a). Although TM topology algorithms predict IF3 to form a TM helix (Extended Data Fig. 8b), our structures revealed that it actually forms a bent helix parallel to the membrane, as suggested previously for Chs3 in yeast[24]. TM5 is an extraordinarily long helix (approximately 80 Å in length) that spans from the TM domain to the cytosolic region and projects into the opposite protomer like a sword (Fig. 1b).

The NTD comprises three subdomains: a microtubule interacting and trafficking (MIT) subdomain, a linkage (LG) subdomain and a swapping (SP) subdomain (Fig. 1c). The MIT subdomain is found in many types of proteins from a wide range of eukaryotes[25,26]. However, within the CHS family, the MIT subdomain has been identified only in those from some

oomycetes[10]. The *Ps*Chs1 MIT subdomain comprises three helices that form an antiparallel helix bundle (Extended Data Fig. 8c). An overlay of the *Ps*Chs1 MIT structure with several other known MIT domains, including those from human, mouse, the oomycete *Saprolegnia monoica* and the fungus *Saccharolobus solfataricus*, reveals significant similarities in the overall topology (Extended Data Fig. 8e). Two conserved noncovalent interhelical interactions stabilize the structure of the MIT subdomain, including an alanine zipper connecting helices α1 and α2 (Extended Data Fig. 8d) and a canonical coiled coil connecting helices α2 and α3. The oomycete MIT subdomain has been previously proposed to be involved in CHS trafficking and targeting to the hyphal tip or in endocytic recycling[27].

The *Ps*Chs1 GT domain, which contains the catalytic machinery, adopts a classical GT-A fold consisting of an eight-stranded β-sheet surrounded by seven α-helices, which tether to the TM region via IF1 through IF3 (Fig. 1c). IF1 leans against TM4 and TM6. IF2 includes the Q(Q/R)XRW motif, which is also conserved among cellulose and hyaluronan synthases[28,29]. IF3 sits on top of IF1 and IF2 and interacts extensively with TM1 through TM4 to stabilize them at the cytosol–lipid interface (Extended Data Fig. 8a).

## Reaction chamber

The reaction chamber in the GT domain includes a tub for substrate binding and a cave for synthetic reactions (Fig. 1d,e). The tub is formed by three previously reported motifs, TMYNE (residues 237–241), DGR (residues 291–293) and DVGT (residues 382–385)[10], along with a newly identified KASKL motif (residues 355–359) (Fig. 1e and Extended Data Fig. 1b). The KASKL motif forms a wall of the tub that separates the tub from the catalytic centre. Specifically, mutations of Asp291 and Asp382, the first two aspartic acids of the signature 'D,D,D,Q(Q/R)XRW' motif, which is conserved among processive β-glycosyltransferases, completely abolished enzyme activity (Extended Data Fig. 2j).

The cave next to the tub is flanked by Glu495 and the catalytic residue Asp496 of the conserved EDR motif (residues 495–497) on one side and by Leu359 of the KASKL motif on the other side (Fig. 1e). The reaction catalysed by CHS is supposed to occur through an S$_N$2 displacement mechanism, with the deprotonated Asp496 acting as the general base that facilitates nucleophilic attack on the anomeric carbon[7]. Consistent with this hypothesis, mutating Asp496 to alanine abolished the catalytic activity and mutating it to asparagine strongly reduced the activity. In addition, mutating Glu495 or Leu359 to alanine resulted in approximately 95% and roughly 70% loss of activity, respectively, suggesting the importance of these residues in catalysis (Extended Data Fig. 2j).

The conserved SWG motif (residues 741–743) is located at a flexible cytoplasmic loop that connects IF3 and TM5. However, the EM map of this loop was weak. This motif might be associated with substrate entrance, as it is located close to the reaction chamber. This possibility is supported by the fact that the mutation of Trp742 to alanine abolished enzyme activity (Extended Data Fig. 2j).

## Chitin-translocating channel

The presumed chitin-translocating channel is located in the TM region and connects the extracellular side of the membrane with the intracellular reaction chamber. The channel is approximately 8–12 Å in width and approximately 40 Å in length (Fig. 1d and Extended Data Fig. 9c). As GlcNAc is approximately 8 Å in width and approximately 5.5 Å in length (Extended Data Fig. 9b), this channel should be able to accommodate a chitin chain containing at least seven GlcNAc units. The entrance of this channel is formed by the VLPGA (residues 452–456), QHFEY (residues 429–433) and QRKRW (residues 535–539) motifs (Fig. 1e). The QHFEY and QRKRW motifs, which belong to IF1 and IF2, respectively, flank the two sides of the channel and interact with each other through a salt bridge formed between Glu432 and Arg536 and hydrophobic interactions between Tyr433 and Trp539. By contrast, the VLPGA motif

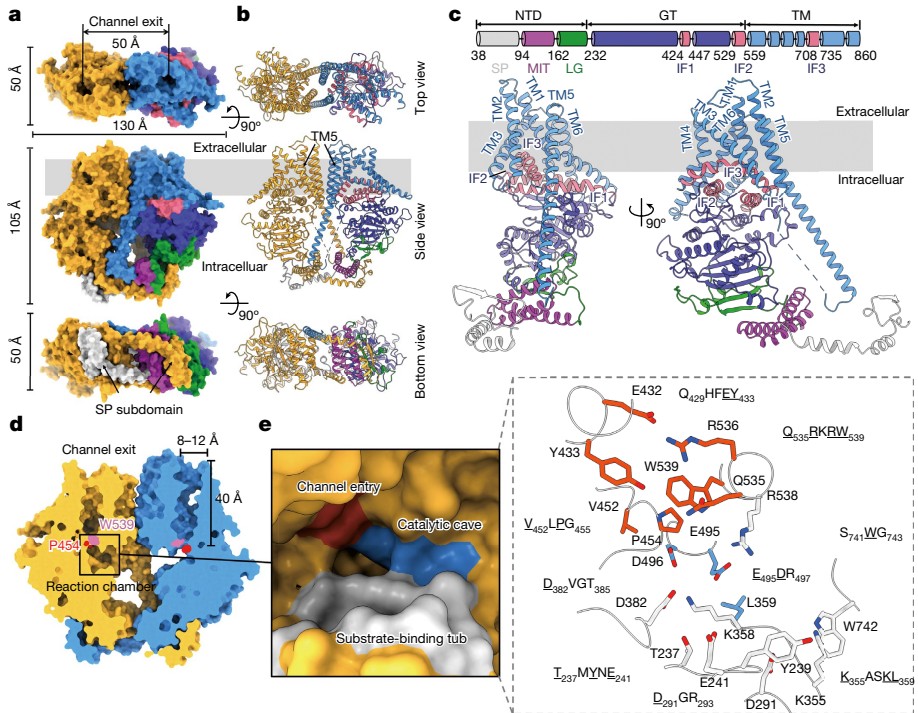

**Fig. 1 | The apo PsChs1 structure. a,b**, The structure of the PsChs1 dimer is shown in surface (**a**) and ribbon (**b**) representations as viewed from the extracellular side of the membrane (top view), within the plane of the membrane (side view), or the cytoplasmic side (bottom view). The approximate position of the membrane is marked with grey shading, and the presumed chitin-translocating channel is marked with arrows. The TM helices, GT domain, IF helices, LG subdomain, MIT subdomain and SP subdomain of one protomer are coloured blue, violet, pink, green, purple and light grey, respectively. The other protomer is coloured yellow. The unresolved region (residues 743–758) is shown as dashed lines. **c**, Domain architecture and ribbon representation of a PsChs1 protomer in two orientations. **d**, Sliced-surface view of the presumed chitin-translocating channel. Pro454 and Trp539 are at the channel entrance and are highlighted in red and pink, respectively. **e**, The reaction chamber of PsChs1 (left) and the conserved motifs that constitute the reaction chamber (right) are shown. The uridine-binding tub, catalytic cave and entrance of the chitin-translocating channel are coloured grey, blue and red, respectively. Residues that are important for enzyme activity are underlined and represented as sticks.

is located on a flexible loop, which we have demonstrated to serve as a gate lock to control access to the chitin-translocating channel (see the following sections for more discussions). The mutation of residues Glu432, Tyr433, Val452, Pro454, Arg536 and Trp539 in those motifs to alanine greatly impaired enzyme activity (Extended Data Fig. 2j).

## The dimeric PsChs1

In a side view, the PsChs1 dimer resembles a hexagonally shaped snowflake (Fig. 1a,b). The two PsChs1 protomers are related to each other by a twofold rotation around an axis perpendicular to the membrane. The PsChs1 dimer buries a large interaction interface, which is stabilized by both the NTD and the TM regions (Fig. 2a).

In the TM region, TM2 of one protomer (PsChs1α) forms numerous hydrophobic interactions with TM5 of the other protomer (PsChs1β), and the loop between TM2 and TM3 of PsChs1α also interacts with TM5 of PsChs1β (Fig. 2b). The cytosolic part of PsChs1β TM5 in turn interacts with the SP, MIT and GT domains of PsChs1α (Fig. 2c). Two NTDs from the two PsChs1 protomers are wrapped by each other to form a symmetric and domain-swapped interface (Fig. 2a). Helix α1 of the SP subdomain is an amphipathic helix with its hydrophobic side facing the MIT subdomain of the opposite protomer, thereby forming hydrophobic interactions with helices α2 and α3 of the MIT subdomain (Fig. 2d). The β1 sheet of the PsChs1α SP subdomain lies parallel to the β1 sheet of the PsChs1β LG subdomain, forming hydrogen bonds and H–π and cation–π stacking interactions with each other (Fig. 2e). Truncation of the SP subdomain (ΔSP) impaired the formation of the PsChs1 dimer and resulted in protein aggregation, but had little effect on enzyme activity (Extended Data Fig. 2l).

The structures of dimeric PsChs1 in complex with the donor substrate or the nascent chitin oligomer suggest that each protomer functions independently but in parallel, that is, two chitin chains are produced concurrently by the two subunits of the CHS homodimer so that these chains assemble with their reducing ends pointing in the same direction. The plot of enzyme activity versus donor substrate concentration fits well to the Michaelis−Menten equation, which gives a Hill coefficient of 1, supporting the absence of cooperative effects between the two PsChs1 protomers (Extended Data Fig. 2e).

## UDP-GlcNAc-bound PsChs1

To obtain the donor substrate UDP-GlcNAc-bound structure, we mutated the catalytically important residue Glu495 to alanine, which allows the enzyme to trap the substrate in its substrate-binding site (Fig. 3a, Extended Data Table 1 and Extended Data Fig. 4). The role of Glu495 is intriguing. In the apo structure, it adopts a conformation different from those in the UDP-bound and UDP/(GlcNAc)$_3$-bound states, with its carboxyl group pointing to the substrate-binding site, which may facilitate the recognition and correct positioning of the donor substrate at the substrate-binding site (Fig. 3b,e). A few previously published substrate-bound GT-A glycosyltransferase structures have shown strong interactions of this glutamate with the sugar moiety of the donor substrate (Extended Data Fig. 8h), supporting that this residue could be important for the recognition and binding of the donor sugar to the substrate-binding site.

The uridine moiety of UDP-GlcNAc is fixed within the tub of the reaction chamber, which consists of a loop region from Thr237 to Glu241 in the GT domain of the enzyme, through a series of interactions with

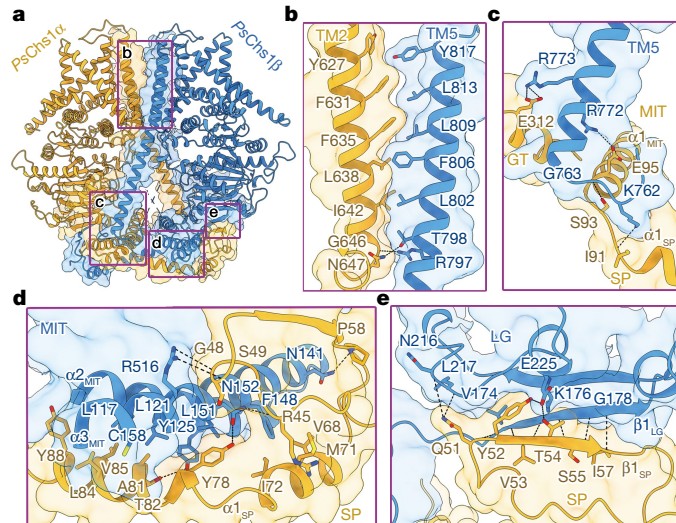

**Fig. 2 | Interface interactions within a *Ps*Chs1 dimer. a**, Overall view of the dimerization interface of *Ps*Chs1 depicted as a ribbon diagram. One protomer (*Ps*Chs1α) is shown in yellow and the other (*Ps*Chs1β) is shown in blue. The dimerization interface is shown in surface view. **b–e**, The four regions involved in interface interactions are highlighted: *Ps*Chs1α TM2 with *Ps*Chs1β TM5 (**b**), *Ps*Chs1α GT domain, MIT subdomain and SP subdomain with *Ps*Chs1β TM5 (**c**), *Ps*Chs1α SP subdomain with *Ps*Chs1β MIT subdomain (**d**) and *Ps*Chs1α SP subdomain with *Ps*Chs1β LG subdomain (**e**). The side chains of residues that are critical for dimerization are shown as sticks. The hydrogen bonds are labelled with black dashes.

the residues in this region (Fig. 3a). The GlcNAc moiety appears to be solvent-exposed and oriented away from the catalytic site. Superposition of UDP-GlcNAc-bound *Ps*Chs1 with apo-*Ps*Chs1 showed no significant local or global structural differences (root mean square deviation of 0.54 Å for 693 Cα atoms). An N-terminal sequence (residues 23–39), unresolved in the *Ps*Chs1 apo structure, appeared as an α-helix in the UDP-GlcNAc complex (Extended Data Fig. 4e). This helix extends into the opposite protomer and interacts with the LG subdomain through hydrogen bonds (Extended Data Fig. 8g). Structural comparison between the apo and UDP-GlcNAc-bound states revealed that residues Pro38, Leu39 and Pro40 are in different locations, indicating that the N-terminal extension of the SP subdomain in the UDP-GlcNAc-bound state adopts a different conformation (Extended Data Fig. 8f).

## UDP/(GlcNAc)₃-bound *Ps*Chs1

The structure of dimeric *Ps*Chs1 bound with a pre-translocating (GlcNAc)$_3$ and a UDP in each *Ps*Chs1 protomer was obtained (Fig. 3c, Extended Data Table 1 and Extended Data Fig. 5). This reflects the post-glycosyl transfer state during chitin biosynthesis. (GlcNAc)$_3$ is trapped within the chitin-translocating channel through a series of interactions, including hydrogen bonds and hydrophobic interactions (Fig. 3c). The second Glc-NAc moiety resides at the entrance of the channel, sandwiched between Pro454 and Trp539. The newly added GlcNAc moiety extends outside the channel, representing a pre-translocation state. As in the substrate-bound *Ps*Chs1, the uridine moiety of UDP is fixed in the uridine-binding tub. Unlike the apo structure, however, an approximately 7 Å retraction of the VLPGA loop (452–458) and an approximately 100° flip of the side chain of Pro454 are observed in the UDP/(GlcNAc)$_3$-bound structure (Fig. 3b, left panel). As a result, the entrance of the chitin-translocating channel is opened. In this state, the side chain of the functionally important residue Glu495 is oriented away from the substrate, similar to that in the UDP-bound or UDP/(GlcNAc)$_3$-bound states (Fig. 3b, right panel), representing a conformation in which the transfer of a sugar moiety

from the donor substrate to the acceptor has just finished. Arg538 of the QRKRW motif is at the entrance of the reaction chamber, and its side chain is highly flexible and adopts different rotamers in apo and substrate-bound states (Fig. 3b, right panel). It forms charge interactions with the diphosphate group of either UDP-GlcNAc or UDP in all the substrate-bound, product-bound and UDP-bound structures. This suggests the role of Arg538 as a guide that directs the donor substrate to enter the reaction chamber and ensures the correct positioning of the substrate for chitin synthesis.

*Ps*Chs1 contains conserved active site residues similar to those previously characterized in GT2 enzymes (in particular, cellulose synthases), such as the *Rs*BcsA monomer from *Rhodobacter sphaeroides* and *Ptt*CesA8 homotrimer from *Populus tremula × tremuloides*[29,30]. However, *Ps*Chs1 has a larger catalytic chamber and a wider translocating channel with smaller surrounding residues than *Rs*BcsA, which allows *Ps*Chs1 to accommodate the bulky *N*-acetyl group of GlcNAc (Extended Data Fig. 9b,c). *Ps*Chs1 residues Thr385 and Leu412, which flank the substrate-binding tub and reside immediately below the entrance of the chitin-translocating channel, are replaced by bulky histidine residues in *Rs*BcsA. Another small residue, Ser357, which belongs to the KASKL motif that borders the substrate-binding tub in *Ps*Chs1, is replaced by His224 in *Rs*BcsA (Extended Data Fig. 9a,e). Mutation of these residues in *Ps*Chs1 to histidine decreases the activity of this enzyme (Extended Data Fig. 2j). Structural comparison shows that *Ps*Chs1 has an α-helix (residues 496–504) and a partially disordered loop (residues 736–759) located in positions similar to those of the 'finger helix' and 'gating loop' of *Rs*BcsA (Extended Data Fig. 9a), which undergo large conformational changes during the translocation of nascent cellulose in *Rs*BcsA[31]. However, our structures showed that neither binding to the donor substrate nor binding to the nascent chitin chain induces significant conformational changes in the finger helix of *Ps*Chs1. The corresponding gating loop in *Ps*Chs1 is only partially resolved, which makes it difficult to discern any conformational changes. Whether these two structural elements have comparable functions in CHSs requires further investigation.

## Mn²⁺ preference for chitin synthesis

In the UDP/(GlcNAc)$_3$-bound structure, the Mn$^{2+}$ ion forms coordinate bonds with the β-phosphate group of the leaving group UDP with a bond length of 2.2–2.3 Å (Fig. 3c), which is consistent with the range of coordinate bonds between a first transition metal and a nitrogen-containing or oxygen-containing compound. The Mn$^{2+}$ ion in the UDP-GlcNAc-bound structure only forms charge interaction with the phosphate groups, with a bond distance of more than 3.5 Å, which is much weaker than a coordinate bond. This may represent the different divalent ion-binding modes in different chitin biosynthesis states (post-catalysis and pre-catalysis), and confirms that the metal ion has an important role in assisting the transfer of the sugar moiety from the donor substrate to a receptor in CHSs and other glycosyltransferases[7].

Mn$^{2+}$ ion is a transition metal with empty 3d electron orbitals and a strong Lewis acid for coordinate bond formation with a phosphate group. Mg$^{2+}$ ion, however, is an alkaline earth metal, which has no empty d orbital, and a weak Lewis acid. Therefore, Mg$^{2+}$ ion can only form charge interactions with a phosphate group, which are weaker than coordinate bonds. The ability of a Mg$^{2+}$ ion in assisting catalysed chitin biosynthesis, therefore, is not as strong as that of a Mn$^{2+}$ ion (Extended Data Fig. 2d).

## A swinging loop directs chitin synthesis

The UDP-bound dimeric *Ps*Chs1 structure represents its nascent chitin-released state (Fig. 3d, Extended Data Table 1 and Extended Data Fig. 6). Compared with *Ps*Chs1 in complex with pre-translocating (Glc-NAc)$_3$ and UDP, the β-phosphate group in the UDP-bound structure is flipped by approximately 55° towards the centre of the reaction chamber, representing a post-synthesis state of the enzyme (Fig. 3c–e). The

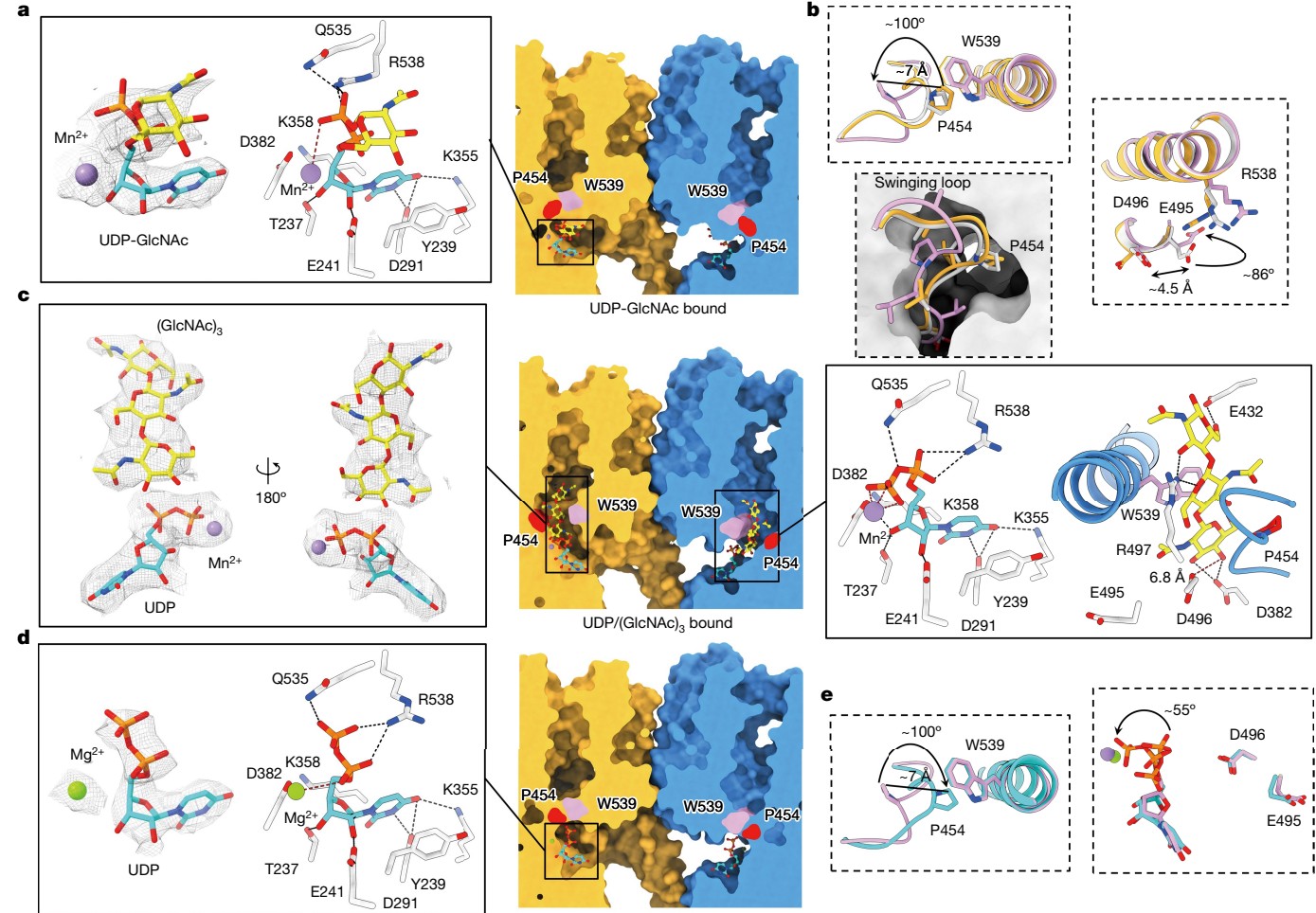

**Fig. 3 | Interactions between *Ps*Chs1 and ligands. a**, Sliced-surface view (right) of the UDP-GlcNAc-binding site (right) and detailed interactions between UDP-GlcNAc and *Ps*Chs1 (left). The residues involved in interactions are labelled and shown as sticks. The hydrogen bonds are labelled as black dashed lines. The interaction between the manganese ion and the ligand is labelled with red dashed lines. The density of the ligand is shown as a grey mesh. **b**, Superimposition of the structures of apo-*Ps*Chs1 (light grey), UDP-GlcNAc-bound *Ps*Chs1 (yellow) and UDP/(GlcNAc)₃-bound *Ps*Chs1 (pink) revealed conformational changes in the swinging loop and catalytic residues.

**c**, Sliced-surface view of the two channels of *Ps*Chs1 in the UDP/(GlcNAc)₃-bound state (middle). The density of UDP/(GlcNAc)₃ (left) and detailed interactions with *Ps*Chs1 (right) are also shown. The hydrogen bonds are labelled as black dashed lines. **d**, Sliced-surface view (right) of the UDP-binding site and detailed interactions between UDP and *Ps*Chs1 (left). The hydrogen bonds are labelled as black dashed lines. **e**, Superimposition of the structures of UDP/(GlcNAc)₃-bound *Ps*Chs1 (pink) and UDP-bound *Ps*Chs1 (cyan).

VLPGA loop (residues 452–456) in the UDP-bound structure moves 7 Å back to its location in the apo and UDP-GlcNAc-bound states, which is different from that in the UDP/(GlcNAc)₃-bound state (Fig. 3b,e). This loop, specifically Pro454, not only functions as a gate to the channel but also stabilizes the second sugar of the nascent chitin oligomer. Mutating Pro454 to alanine abolished the enzyme activity (Extended Data Fig. 2j). The position of the VLPGA loop in different states of the enzyme suggests that this loop serves as a 'gate lock', which prevents the donor substrate from leaving before being linked to a growing chitin oligomer. In addition, it directs the head of the product polymer through the exit of the channel towards the extracellular side of the cell membrane.

The apo structure of *Ps*Chs1 is the first example of a membrane-integrated processive glycosyltransferase with an apparently continuous but empty transmembrane channel, providing a real off-state of the enzyme before chitin synthesis is initiated. Molecular dynamics simulations of the apo and UDP/(GlcNAc)₃-bound states of *Ps*Chs1 suggested that the VLPGA motif also acts as a permeability barrier that prevents water flux across the membrane in the apo state (Extended Data Fig. 9f,g and Supplementary Videos 1 and 2).

This swinging loop appears to be highly conserved in CHSs because all chitin synthases contain a similar loop in the channel with the signature Pro454 residue (Extended Data Fig. 1b). It is interesting that *Rs*BcsA contains an 'FFCGS' motif at the corresponding location (Extended Data Fig. 9d,e). Although its sequence shows some level of conservation with the VLPGA motif of CHSs, it is noticeable that the cysteine in this motif is different from the signature proline in the VLPGA motif, which can adopt specific conformations different from those of cysteine or other non-proline residues. The FFCGS motif in BcsA, therefore, may not function in the same mode as the VLPGA gate lock loop in CHSs[29].

## Chitin biosynthesis inhibition by NikZ

The cryo-EM structure of *Ps*Chs1 in complex with NikZ, which competitively inhibits *Ps*Chs1 activity by a inhibition constant ($K_i$) value of 151.1 ± 4.8 μM, was resolved (Fig. 4, Extended Data Table 1 and Extended Data Fig. 7). As expected for a substrate analogue, NikZ binds to the uridine-binding tub through its uridine moiety in a manner identical

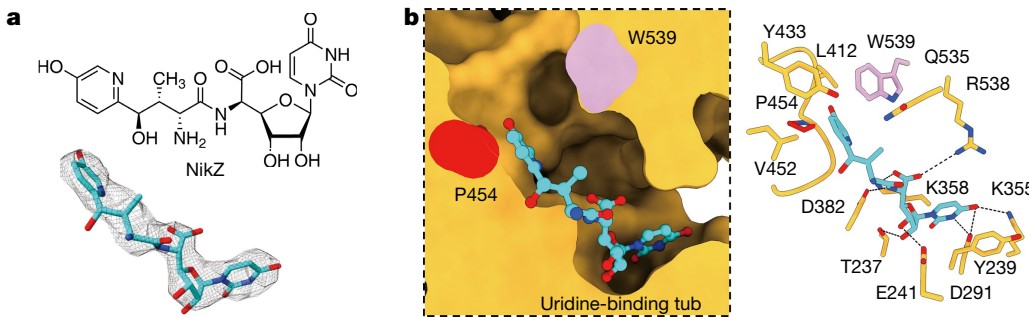

**Fig. 4 | Inhibition of *Ps*Chs1 by NikZ. a**, Chemical structure and EM density of NikZ. **b**, Sliced-surface view (left) of the NikZ-binding site and detailed interactions between NikZ and *Ps*Chs1 (right). The hydrogen bonds are labelled as black dashed lines.

to that of UDP-GlcNAc, thus blocking the binding of a donor substrate to this position (Figs. 3a and 4). Moreover, the hydroxypyridine moiety of NikZ occupies a large part of the reaction chamber as well as the entrance of the translocating channel, thereby further blocking the donor substrate from entering the reaction chamber for chitin biosynthesis (Fig. 4b). The hydroxypyridine ring is positioned at the entrance of the translocating channel and mimics a pre-translocating sugar unit to induce the opening of the gate lock-controlled channel. It hydrophobically interacts with the residues around the channel entrance, particularly Tyr433 from the QHFEY motif, Val452 and Pro454 from the VLPGA gate lock loop, and Trp539 from the QRKRW motif, as well as Leu412 of the β-strand near the gate lock loop. Mutation of each of these residues not only reduced the inhibitory activity of NikZ but also significantly impaired enzyme activity (Extended Data Fig. 2j,k).

## Discussion

*Ps*Chs1 exhibits a unique quaternary structure composed of two mirror-imaged protomers that are assembled into an N-terminally intertwined dimer, suggesting a parallel and directional mode of chitin biosynthesis. The formation of dimeric and oligomeric CHS complexes has been previously reported in vitro and in vivo[24,32,33] and is in agreement with the structural data of *Ps*Chs1 in this study and the recently published *Ca*Chs2 from *Candida albicans*[34]. Both dimeric *Ps*Chs1 and *Ca*Chs2 produce α-chitin. Sequence alignment between *Ps*Chs1 and *Ca*Chs2 or other CHSs indicates that the dimeric interface residues are highly variable (Extended Data Fig. 1a) and that the N-terminal SP and MIT subdomains, which are important for *Ps*Chs1 dimerization, are present only in oomycete CHSs[10]. These observations suggest that the dimerization mechanism shown in our structures may be unique to CHSs from oomycetes and some fungi, and different oligomerization strategies may have evolved for CHSs from other taxa. Nonetheless, CHS oligomerization appears to be important for the formation of chitin fibrils because the well-ordered assembly of single catalytic units is related to proper alignment of nascent sugar chains before their coalescence and the formation of chitin nanofibrils[3].

We noticed that chitin biosynthesis was initiated in the absence of free GlcNAc (Extended Data Fig. 2d,f), which suggested that free GlcNAc is not required for biosynthesis and that the first molecule that binds to the active site in the absence of free GlcNAc is UDP-GlcNAc. The spatial restrictions in the substrate-binding pocket together with the nonreducing end-chain elongation mechanism of CHS[35] exclude the possibility that two UDP-GlcNAc molecules bind simultaneously to the active site to initiate the reaction. Although GlcNAc is not a positive effector for *Ps*Chs1 (Extended Data Fig. 2e), the addition of free GlcNAc instead of chito-oligosaccharides could speed up the synthesis

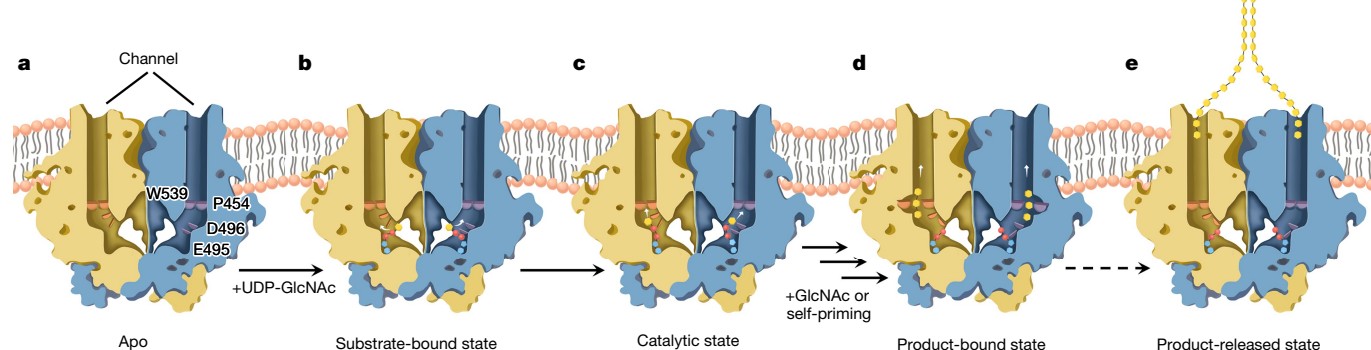

**Fig. 5 | A model of chitin biosynthesis. a**, In the apo enzyme, the entrance of the chitin-translocating channel is blocked by the gate lock loop. **b**, Chitin synthesis starts when the donor substrate UDP-GlcNAc enters the reaction chamber and resides in the uridine-binding tub. The white arrows indicate the moving direction of the GlcNAc moiety. **c,d**, A GlcNAc unit produced from UDP-GlcNAc hydrolysis (self-priming mechanism) or an exogenously added GlcNAc is proposed to act as an acceptor to initiate chitin biosynthesis. This process should include the following steps: Glu495 binds and stabilizes the donor substrate, and the catalytic residue Asp496 interacts with the acceptor GlcNAc and assists the nucleophilic attack on a donor substrate; a divalent metal ion binds to the diphosphate group of the donor substrate and helps in the release of the UDP moiety to complete the formation of a disaccharide. It is

likely that disaccharide formation induces conformational changes in the gate lock loop and the flipping of Pro454, thereby opening the entrance and allowing access of the nascent sugar chain to the translocating channel. When the addition of a sugar unit is completed, the catalytic Glu495 and the metal ion-bound diphosphate of the leaving UDP rotate away from the reaction centre (**d**). The white arrows in **c** and **d** indicate the moving directions of the products. **e**, After many rounds of reaction (dashed arrow), the enzyme adopts a post-synthesis state and a chitooligosaccharide product is discharged: the translocating channel is closed by the gate lock, the leaving group UDP sitting at the substrate-binding site needs to be replaced by a new donor substrate, and the positions of Glu495 and the catalytic residue Asp496 are restored to their pre-synthesis states for a new cycle of chitin biosynthesis to be initiated.

(Extended Data Fig. 2f). These results together suggest that GlcNAc is an acceptor for the initial step of chitin biosynthesis. Similarly, a 'self-priming' mechanism has been proposed for yeast chitin synthase[36] and cellulose synthase[37,38] and has very recently been suggested for a viral homologue of hyaluronan synthase[39].

Our structural data and previously published cellulose synthase structures have shown that each promoter in a homodimeric or oligomeric enzyme synthesizes and translocates a single sugar chain independently and in parallel. Recent structural insights into cellulose biosynthesis explained that the 180° alternating arrangement of the sugar units can be achieved by a simple rotation of the terminal sugar unit around the glycosidic bond, eliminating the need for a dual substrate-binding site[29,31].

It is clear that chitin polymer elongation is achieved by repeated steps of adding a sugar moiety from the donor substrate to the acceptor substrate through an $S_N2$ displacement reaction. The elongated chitin polymer is supposed to be discharged from the head end through the translocating channel to the extracellular side of the transmembrane domain. Our *Ps*Chs1 structures revealed that the entrance to this channel is blocked by the swinging loop, specifically residue Pro454, in the apo and substrate-bound states, but it is open in the (GlcNAc)$_3$-bound state. Thus, the swinging (VLPGA) loop may function as a gate lock that facilitates the directed transport of nascent chitin across the cell membrane (Fig. 3b,e). With the gate lock loop being closed, the space around the catalytic centre and the channel entrance allows only a single unit GlcNAc acceptor to be positioned. A large conformational change is needed in the VLPGA loop to open the gate lock and to allow access of the nascent disaccharide to the channel when a GlcNAc unit joins the acceptor GlcNAc (exogenously added or derived from UDP-GlcNAc) to form a GlcNAc dimer. The gate lock is supposed to be kept open during chain elongation until biosynthesis is completed and the chitin chain is discharged.

Putting together our five *Ps*Chs1 structures and biochemical data, we propose a model of chitin biosynthesis, as shown in Fig. 5. Furthermore, the synthesized chitin chains might be aligned in parallel along the *a* axis (perpendicular to the pyranose ring) through the hydrophobic surface of their pyranose rings to form chitin sheets, which may be self-assembled in an antiparallel manner along the *b* axis (parallel to the pyranose ring) to finally form α-chitin (Extended Data Fig. 9h), the more stable chitin allomorph[40]. The dimeric *Ps*Chs1 is consistent with the formation of chitin sheets by the parallel arrangement of single chains.

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

# Methods

## Protein expression and purification

The codon-optimized cDNA of PsChs1 was synthesized (GenScript) and ligated into the pcDNA3.1 vector in C-terminal fusion with tandem twin Strep tag and Flag tag. For PsChs1 overexpression, HEK293F cells (Invitrogen) were cultured in SMM 293T-II medium (Sino Biological Inc.) at 37 °C in the presence of 5% $CO_2$. When the cell density reached $1.5–2.0 \times 10^6$ cells per ml, a mixture of recombinant plasmids and poly-ethylenimine 40000 (Polysciences) at a ratio of 1:3 was added to the cell culture to transiently overexpress PsChs1.

Cells were harvested by centrifugation at $1,000\,g$ after transfection for 48 h and resuspended in lysis buffer (25 mM Tris-HCl pH 8.0 and 150 mM NaCl) supplemented with protease inhibitor cocktail (Topscience) including 2.6 µg ml$^{-1}$ aprotinin, 1.4 µg ml$^{-1}$ pepstatin, 10 µg ml$^{-1}$ leupeptin hemisulfate and 1 mM PMSF. Cells were lysed through a chilled high-pressure homogenizer (ATS) and centrifuged at $16,000\,g$ for 30 min at 4 °C. The supernatant was ultracentrifuged at $150,000\,g$ for 1 h at 4 °C to collect the membranes. The membranes were solubilized in lysis buffer supplemented with 1% digitonin (Biosynth) and the protease inhibitor cocktail. After incubation at 4 °C for 2 h, the solution was ultracentrifuged at $150,000\,g$ for 30 min at 4 °C, and the supernatant was loaded onto a preequilibrated Strep-TactinXT 4Flow cartridge (IBA Lifesciences). The column was washed with buffer W containing 25 mM Tris-HCl pH 8.0, 150 mM NaCl, 0.05% digitonin and the protease inhibitor cocktail. The target proteins were eluted with buffer W supplemented with 50 mM biotin. The eluate was then loaded onto anti-Flag M2 affinity resin (Sigma). The target protein was eluted with buffer W supplemented with 200 mg ml$^{-1}$ Flag peptide. The sample was concentrated using a 100-kDa cut-off Centricon (Millipore) and subjected to size-exclusion chromatography on a Superdex 200 Increase 10/300 GL column (GE Healthcare) equilibrated with buffer W. The peak fractions were analysed by SDS−PAGE. The expression and purification procedures for mutated or truncated recombinant proteins were the same as those conducted with the native PsChs1. The expression of mutated or truncated recombinant proteins was confirmed by western blot analysis using an anti-Flag M2 antibody (1:1,000 dilution; Sigma). For gel source data, see Supplementary Fig. 1.

## Activity assays

The chitin synthase activity assay was performed according to published procedures with slight modifications[41]. The microtitre plates were coated with wheat germ agglutinin (50 µg ml$^{-1}$; Sigma) and blocked with bovine serum albumin blocking buffer. The plates were stored at −20 °C for further assays. The plates were emptied by centrifugation before usage. Next, 50 µl of the reaction mixture (1 mM UDP-GlcNAc, 50 mM Tris-HCl buffer pH 7.5 and 150 mM NaCl) was added to each well followed by 50 µl of enzyme suspension to a final volume of 100 µl. After incubation at 30 °C for 1 h, the plates were emptied and washed three times. Then, 100 µl of wheat germ agglutinin−horseradish peroxidase (0.5 µg ml$^{-1}$; Sigma) was added, and the mixture was incubated for 30 min at 30 °C. The plate was emptied through centrifugation and washed three times. Of peroxidase substrate reagent, 100 µl was added, and absorbance at 652 nm was immediately determined for 5 min. The GlcNAc content and chitin synthase activity after each treatment were calculated by using a standard curve, which was prepared following the same procedure as previously described[42]. The specific enzyme activity was expressed as nmol GlcNAc per µg protein per hour. Each test was repeated three times.

For measuring enzyme kinetics, the reaction mixtures containing 1 µg of PsChs1, 4.5 µM to 10 mM UDP-GlcNAc, 50 mM Tris-HCl buffer (pH 7.5) and 150 mM NaCl were incubated in a final volume of 100 µl in the absence or presence of 1 mM GlcNAc. The data were analysed by GraphPad Prism software.

## Characterization of synthesized chitin

To visualize synthesized chitin in vitro through a scanning electron microscope, 10 µl of the reaction sample was applied onto a silicon support ($5 \times 5$ mm) and dried at room temperature to fix the sample. The silicon supports containing the samples were washed with distilled water five times to remove salt and sputter-coated with gold-palladium. Samples were observed using a Regulus 8100 microscope (Hitachi) at 10 kV.

For confocal laser scanning microscopy, reaction samples were incubated with wheat germ agglutinin coupled to fluorescein isothiocyanate (0.02 mg ml$^{-1}$; Genetex) for 15 min. After washing five times with distilled water, the samples were imaged for fluorescein isothiocyanate fluorescence (absorption at 490 nm; emission at 520 nm) using an LSM 880 laser scanning confocal microscope and appropriate filter sets (Zeiss).

ATR-FTIR spectra and X-ray diffraction were used to determine the crystal isomorph of synthesized chitin. Chitin derived from shrimp and chitin derived from Satsuma tubeworm were used as reference samples for α-chitin and β-chitin, respectively. FTIR spectra were obtained using a Nicolet iS5 FTIR Spectrometer (Thermo Fisher Scientific) with a diamond ATR unit over the frequency range of 4,000 to 600 cm$^{-1}$ in absorbance mode. Spectroscopy was recorded at a resolution of 4 cm$^{-1}$ and 128 scans. The data were analysed with Omnic software. X-ray diffraction data were obtained at 40 kV, 30 mA and 2θ with a scan angle from 5° to 40° using a Rigaku D max 2000 system (Rigaku) at the Institute of Chemistry, Chinese Academy of Sciences. The data were analysed with Jade software.

## EM sample preparation and data collection

All cryo-EM grids were prepared by loading 3 µl of protein at a concentration of approximately 5 mg ml$^{-1}$ onto glow-discharged holey carbon grids (Au R1.2/1.3, 300-mesh; Quantifoil or Beijing EBO Technology Limited). Grids were blotted for 6−8 s at 4 °C and 100% humidity and plunged into liquid ethane cooled by liquid nitrogen using Vitrobot Mark IV (FEI). The ligand-bound PsChs1 complexes were prepared by incubating the protein with 0.5 mM UDP-GlcNAc or 5 mM NikZ before sample preparation. Grids were screened and checked using a 200 kV Tecnai G2 F20 TWIN TMP microscope (FEI) in the State Key Laboratory of Membrane Biology, Institute of Zoology, Chinese Academy of Science.

Data collection for apo-bound, UDP-GlcNAc-bound, UDP/(GlcNAc)₃-bound and NikZ-bound PsChs1 was performed using a 300 kV Titan Krios microscope (FEI) equipped with a Gatan K3 direct detector by SerialEM in the Center for Biological Imaging, Institute of Biophysics, Chinese Academy of Science. Images were recorded in super-resolution mode by beam-image shift data collection methods[43] at a magnification of ×22,500, resulting in a physical pixel size of 1.07 Å. The exposure time for each stack of 32 frames was 3.43 s, corresponding to a total dose of approximately 60 e$^-$ Å$^{-2}$ and a defocus ranging from −1.2 to −2.0 µm. Data collection for UDP-bound PsChs1 was performed using a 300 kV Titan Krios microscope equipped with a Gatan K2 direct detector at a magnification of ×130,000 by SerialEM in the Cryo-Electron Microscopy Research Center, Southern University of Science and Technology. Images were automatically acquired using the same conditions as those used for apo-PsChs1.

## Cryo-EM data processing

For the PsChs1 apo structure, the output movie stacks were subjected to beam-induced motion correction and dose-weighting using MotionCor2 (ref. [44]). Contrast transfer function parameters on each summed image were estimated with the Gctf program[45]. We selected a subset of the particles using the Laplacian-of-Gaussian method, processed with reference-free 2D classification, and the five 2D classes were selected as references for automatic particle picking of the complete dataset

of 3,341 images. This resulted in a total of 2,379,125 particles. After one round of reference-free 2D classification, 927,432 particles were selected for an additional two rounds of 3D classification. Particles were classified into five classes using the initial model obtained from the 3D initial model in 3D classification with C2 symmetry. A class with more complete N-terminal regions was selected with 161,907 particles for 3D auto-refinement, contrast transfer function refinement and Bayesian polishing, which resulted in a 3.3 Å density map. All the processing steps were conducted in RELION 3.08 (ref. [46]) as shown in Extended Data Fig. 3.

For the UDP-GlcNAc-bound PsChs1 complex, a total of 1,855,418 particles were selected in automatic particle picking from a total of 3,878 images. A total of 1,811,918 particles were used in two rounds of 3D classification with C1 and C2 symmetry, using the native density as the initial model. The class with more complete N-terminal regions was selected with 350,132 particles for 3D auto-refinement, contrast transfer function refinement and Bayesian polishing, which resulted in a 3.3 Å density map. The processing steps for UDP/(GlcNAc)$_3$-bound, UDP-bound and NikZ-bound PsChs1 complexes were the same as those for the UDP-GlcNAc-bound PsChs1 complex. All the processing steps are shown in Extended Data Figs. 4–7.

## Model building and refinement

The de novo model building of PsChs1 was performed based on the BcsA subunit of the cellulose synthase structure (Protein Data Bank code 4HG6), which was initially docked into half of a 3.3 Å resolution cryo-EM map of apo-PsChs1 using UCSF Chimera[47]. The model of one protomer was rebuilt manually based on the cryo-EM density with COOT[48], and the other half was symmetrically docked using this rebuilt model. The dimeric structure was real-space refined using Phenix[49] in C2 symmetry.

For PsChs1 complexes, the models were rebuilt using the apo structure as an initial model and refined following the same procedure. The ligands were modelled into the cryo-EM density map. MolProbity[50] was used to evaluate the geometries of the structures, and the statistical information is listed in Extended Data Table 1. Figures were prepared with UCSF ChimeraX[51].

## Molecular dynamics simulation

The molecular dynamics software package GROMACS v2019.3 was used with the Gromos53a5 force field to compare the structural properties obtained from computational simulation with the structural properties determined from experiments[52]. The experimentally determined structures of apo-PsChs1 (closed conformation) and product-bound PsChs1 (open conformation, with the removal of UDP and (GlcNAc)$_3$ in the channel) were used for the comparative validation study. The simulation cell consists of a phosphatidylethanolamine-phosphatidylglycerol (POPE) bilayer with 598 lipids and 48,555 explicit simple point charge (SPC) solvent molecules and was simulated in NVT (number of particles, system volume and temperature) and NPT (number of particles, system pressure and temperature) ensembles (pressure at 1 atm; temperature at 300 K) using the Parinello–Rahman barostat and Berendsen thermostat (300 K)[53,54]. Electrostatic interactions for long-range electrostatics were calculated using Particle Mesh-Ewald. All bond lengths and the geometry of covalent bonds containing water molecules were constrained using the LINCS[55] and SETTLE[56] algorithms, respectively. A molecular dynamics simulation of 50 ns was carried out to evaluate the water flux through the channel.

## Reporting summary

Further information on research design is available in the Nature Research Reporting Summary linked to this article.

## Data availability

The atomic coordinates and EM map for the apo, UDP-GlcNAc-bound, UDP/(GlcNAc)$_3$-bound, UDP-bound and NikZ-bound PsChs1 have been deposited in the Protein Data Bank (www.rcsb.org) with accession codes 7WJM, 7WJN, 7X05, 7X06 and 7WJO, respectively, and in the Electron Microscopy Data Bank (www.ebi.ac.uk/pdbe/emdb/) with the accession codes EMD-32545, EMD-32546, EMD-32917, EMD-32918 and EMD-32547, respectively.

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

**Acknowledgements** We thank Z. Rao and H. Li for their valuable comments during the preparation of this manuscript; P. Xia for his help on EM sample screening; Y. Huo and T. Jiang for their help on apo-PsChs1 data collection; X. Li, B. Zhu, D. Fan, T. Niu, X. Huang and all staff at the Center for Biological Imaging, Institute of Biophysics, Chinese Academy of Sciences, and S. Xu, X. Ma and the staff at the Cryo-EM Center of Southern University of Science and Technology for their support on single-particle cryo-EM data collection; X. Gao, M. Zhao, Y. Zhu, X. Song, W. Liu and D. Zheng, and the Large-scale Instruments and Equipments Sharing Platform of Beijing University of Technology for their support; and L. Niu, M. Zhang and F. Yang for mass spectrometer analysis. This work was supported by the National Natural Science Foundation of China (31830076, 32161133010 and 31901916), the Shenzhen Science and Technology Program (KQTD20180411143628272), and the Special Funds for Science Technology Innovation and Industrial Development of Shenzhen Dapeng New District (PT202101-02).

**Author contributions** Q.Y. conceived and designed this project. W.C., Y.L. and D.W. prepared the samples and screened the cryo grids. W.C. and A.Y. performed the data acquisition. W.C. and L.C. performed the product characterization. R.S. and Z.Y. performed the molecular dynamics simulation. P.C. and Y.G. performed the image processing and structure determination. Q.Y., W.C. and H.M. wrote the manuscript.

**Competing interests** The authors declare no competing interests.

**Additional information**
**Correspondence and requests for materials** should be addressed to Yong Gong or Qing Yang.

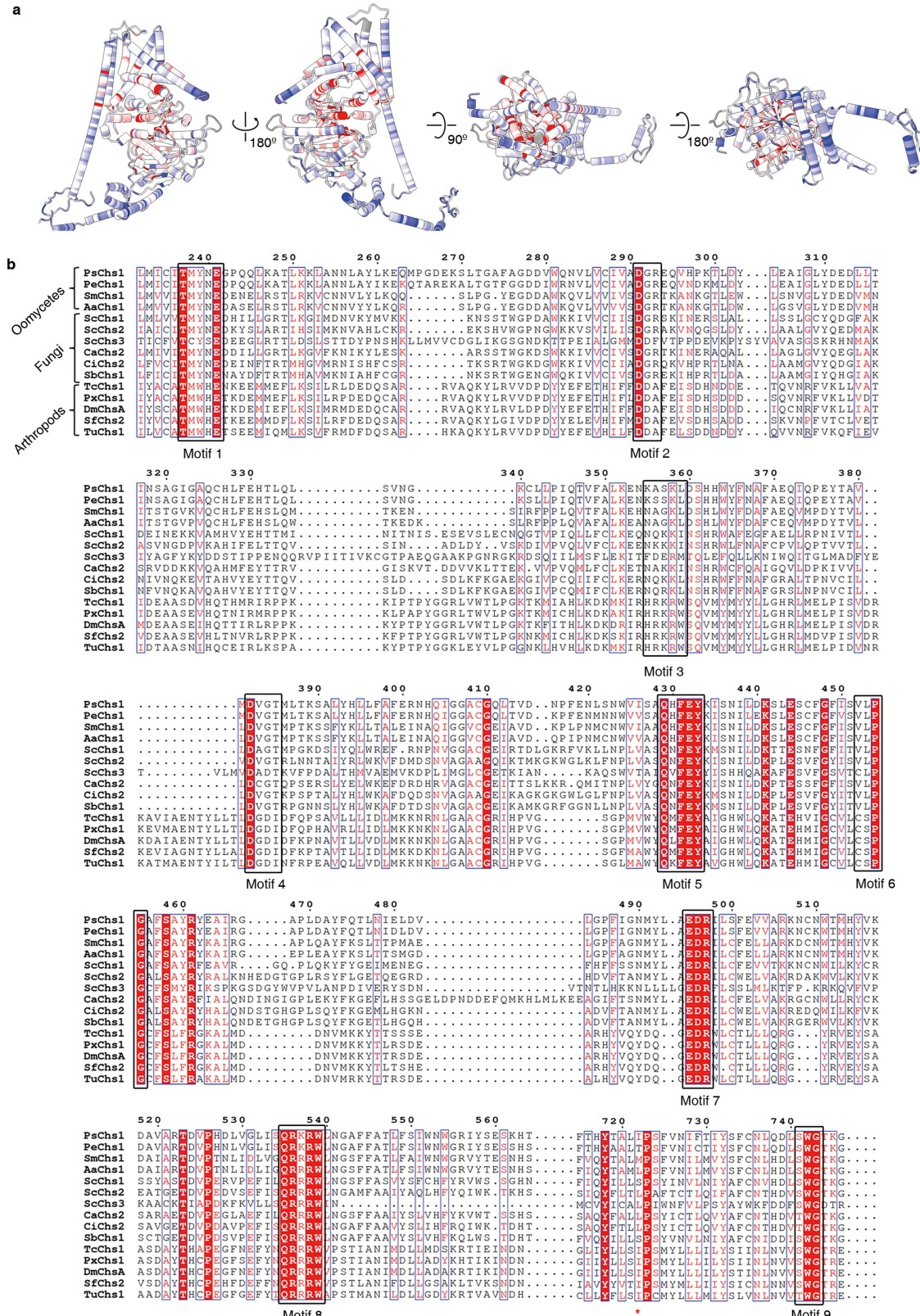

**Extended Data Fig. 1** | See next page for caption.

**Extended Data Fig. 1 | Sequence conservation among CHSs from oomycetes, fungi, and arthropods. a**, The structure of *Ps*Chs1 is colored according to sequence conservation among CHSs from different species. Residues are colored red, white and blue to indicate identical, similar and variable residues, respectively. The following sequences were used: *Ps*Chs1 (XP_009524159) (this study), *Pe*Chs1 (RMX65634) from *Peronospora effusa*, *Sm*Chs1 (ADE62520) from *Saprolegnia monoica*, *Aa*Chs1 (RHZ26641) from *Aphanomyces astaci*, *Sc*Chs1 (NP_014207), *Sc*Chs2 (NP_009594), and *Sc*Chs3 (NP_009579) from *Saccharomyces cerevisiae* (XP_011318411), *Ca*Chs2 (KHC63241) from *Candida albicans*, *Ci*Chs2 (KMP06892) from *Coccidioides immitis*, *Sb*Chs1 from *Sporothrix brasiliensis* (XP_040620288), *Tc*Chs1 from *Tribolium castaneum* (AAQ55059), *Px*Chs1 (BAF47974) from *Plutella xylostella*, *Dm*ChsA (AAG22215) from *Drosophila melanogaster*, *Sf*Chs2 (AAS12599) from *Spodoptera frugiperda*, and *Tu*Chs1 (AFG28412) from *Tetranychus urticae*. **b**, Multiple sequence alignment of the catalytic domain of CHSs from different species was generated using Clustal X. Amino acids are highlighted by white (on red background), red and gray letters based on sequence identity, similarity and variability, respectively. Black boxes refer to conserved motifs, including motif 1 (TMYNE), motif 2 (DGR), motif 3 (KASKL), motif 4 (DVGT), motif 5 (QHFEY), motif 6 (VLPG), motif 7 (EDR), motif 8 (QRKRW), and motif 9 (SWG).

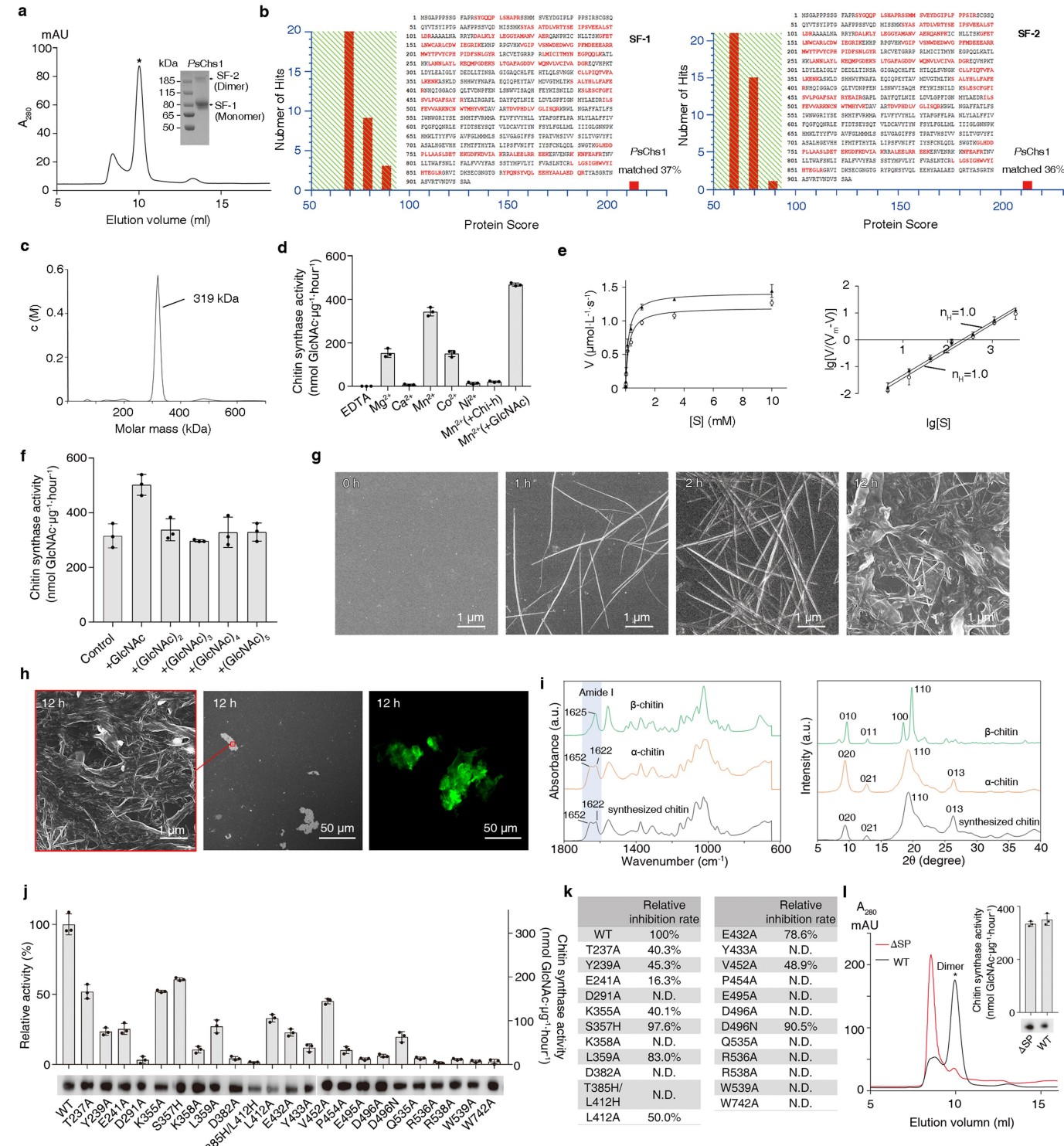

**Extended Data Fig. 2** | See next page for caption.

**Extended Data Fig. 2 | Identification of recombinant *Ps*Chs1. a**, Size exclusion chromatography profile and SDS–PAGE analysis of purified *Ps*Chs1 (inset, right). **b**, The two protein bands (SF-1 and SF-2) in SDS–PAGE were verified by peptide mass fingerprint. The histogram presents the distributions of protein scores reported in Mascot Server (http://www.matrixscience.com/), which uses a two-sided expectation value test without any adjustments for data analysis. Protein scores > 93 are significant (p < 0.05). **c**, The molar mass of recombinant *Ps*Chs1 was determined by analytical ultracentrifugation. The theoretical molar mass of the *Ps*Chs1 monomer is 107 kDa, and the molar mass of the main peak is somewhere between that of the dimer and trimer. Considering that these peaks were obtained in the presence of detergent, *Ps*Chs1 may exist as a dimer in solution. **d**, The graph shows that *Ps*Chs1 activity is affected by metal ions, and the product is degraded by chitinase. The activity is increased by the addition of GlcNAc. Chi: chitinase. **e**, The Michaelis–Menten curve (left) and Hill plot (right) of the reaction catalyzed by *Ps*Chs1 in the presence (▲) or absence (○) of free GlcNAc. Error bars, mean ± SD. **f**, Enzymatic activity of *Ps*Chs1 in the presence of different chitooligosaccharides at a concentration of 1 mM. **g**, Scanning electron microscopy images of *in vitro* synthesized chitin on a time scale of hours. **h**. Scanning electron microscopy and laser confocal microscopy analysis of synthesized chitin aggregates. **i**, Fourier transform infrared spectra (left panel) and X-ray diffraction profiles (right panel) of α-chitin, β-chitin, and the synthesized chitin. **j**, Enzyme activities of wild-type *Ps*Chs1 and various mutants. **k**, Relative inhibition rate of NikZ for different *Ps*Chs1 mutants. N.D.: not determined, the enzyme activity of the mutant was too low to measure the inhibitory activity of NikZ. **l**, Truncation of SP affects dimerization but not enzyme activity. Data in **d**, **e**, **f**, **g**, **l** are presented as the mean ± s.d. from three independent experiments (n = 3). Data in **g**, **h** are repeated independently three times with similar results. The uncropped gel images for **a**, **j**, **l** are available as supplementary information Fig. 1.

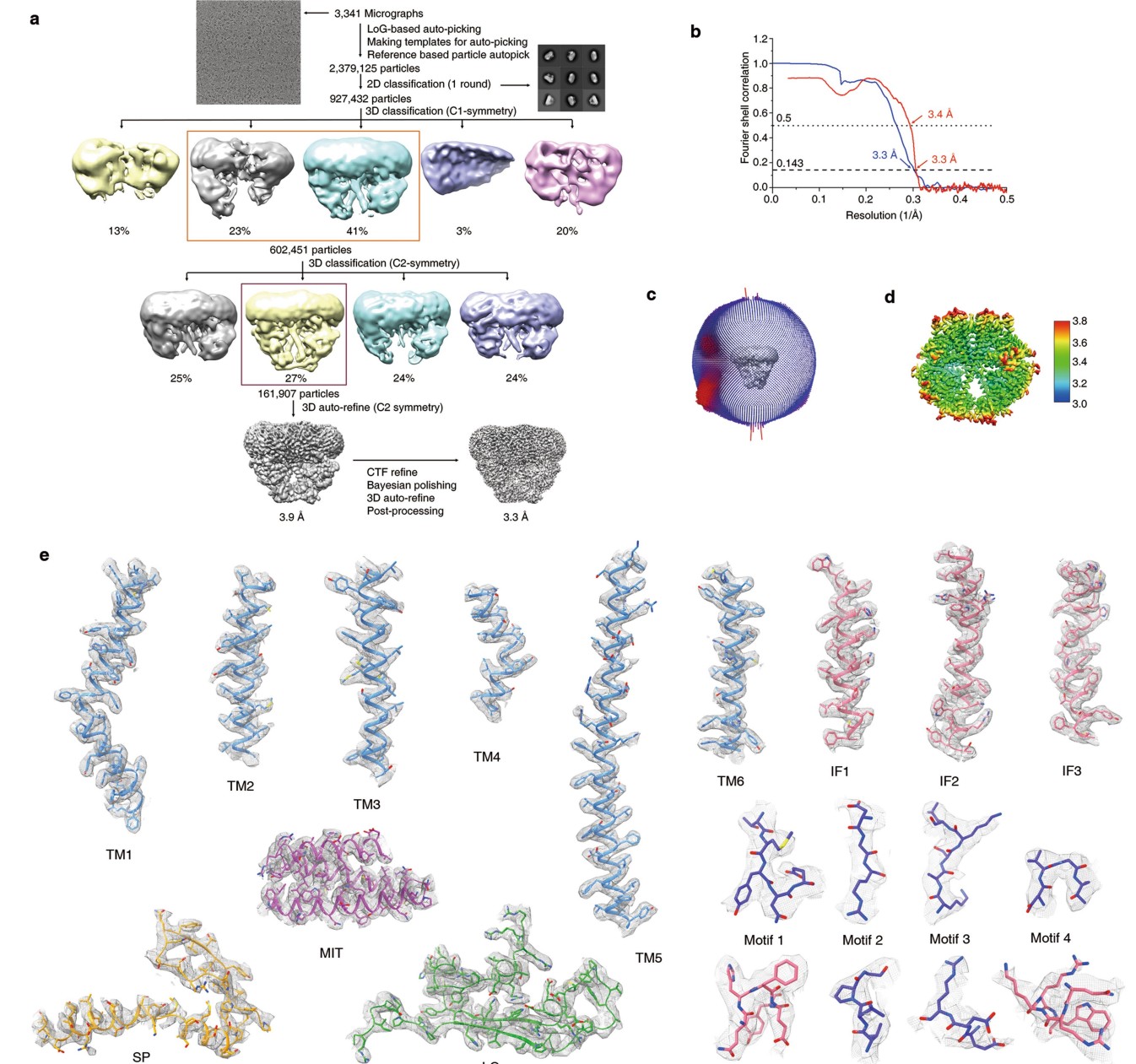

**Extended Data Fig. 3 | Cryo-electron microscopy (Cryo-EM) analysis of apo *Ps*Chs1. a**, Flowchart of cryo-EM data processing for apo-*Ps*Chs1. **b**, Gold-standard Fourier shell correlation (FSC) curves of apo-*Ps*Chs1, including FSC between two independently refined half-maps generated by RELION (blue) and model-to-map FSC generated by Phenix (red). **c**, Angular distribution of the particles contributing to the final reconstruction of *Ps*Chs1. **d**, The map of *Ps*Chs1 is colored according to local resolution estimation. **e**, Sample maps of TM helices, IF helices, N-terminal subdomains, and eight conserved motifs of apo-*Ps*Chs1.

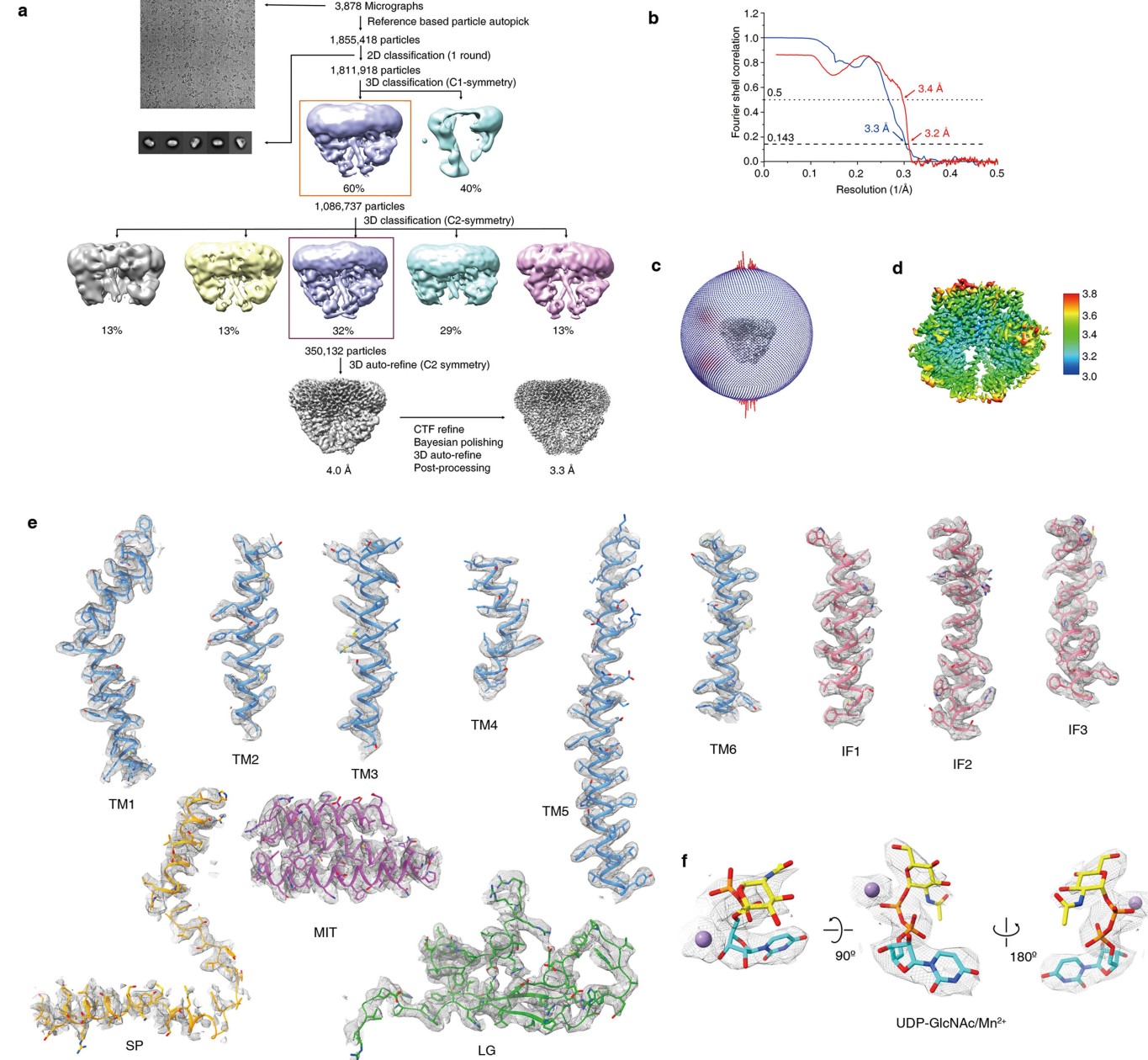

**Extended Data Fig. 4 | Cryo-EM data analysis of UDP-GlcNAc-bound *Ps*Chs1.**
**a**, Flowchart of cryo-EM data processing. **b**, FSC curves of UDP-GlcNAc-bound
*Ps*Chs1, including FSC between two independently refined half-maps
generated by RELION (blue) and model-to-map FSC generated by Phenix (red).
**c**, Angular distribution of the particles contributing to the final reconstruction

of *Ps*Chs1. Each column represents one view, and the size of the column is
proportional to the number of particles in that view. **d**, The map of
UDP-GlcNAc-bound *Ps*Chs1 is colored according to estimated local resolution.
**e**, Sample maps of TM helices, N-terminal subdomains, and IF helices of
UDP-GlcNAc-bound *Ps*Chs1. **f**, EM density of UDP-GlcNAc/Mn²⁺.

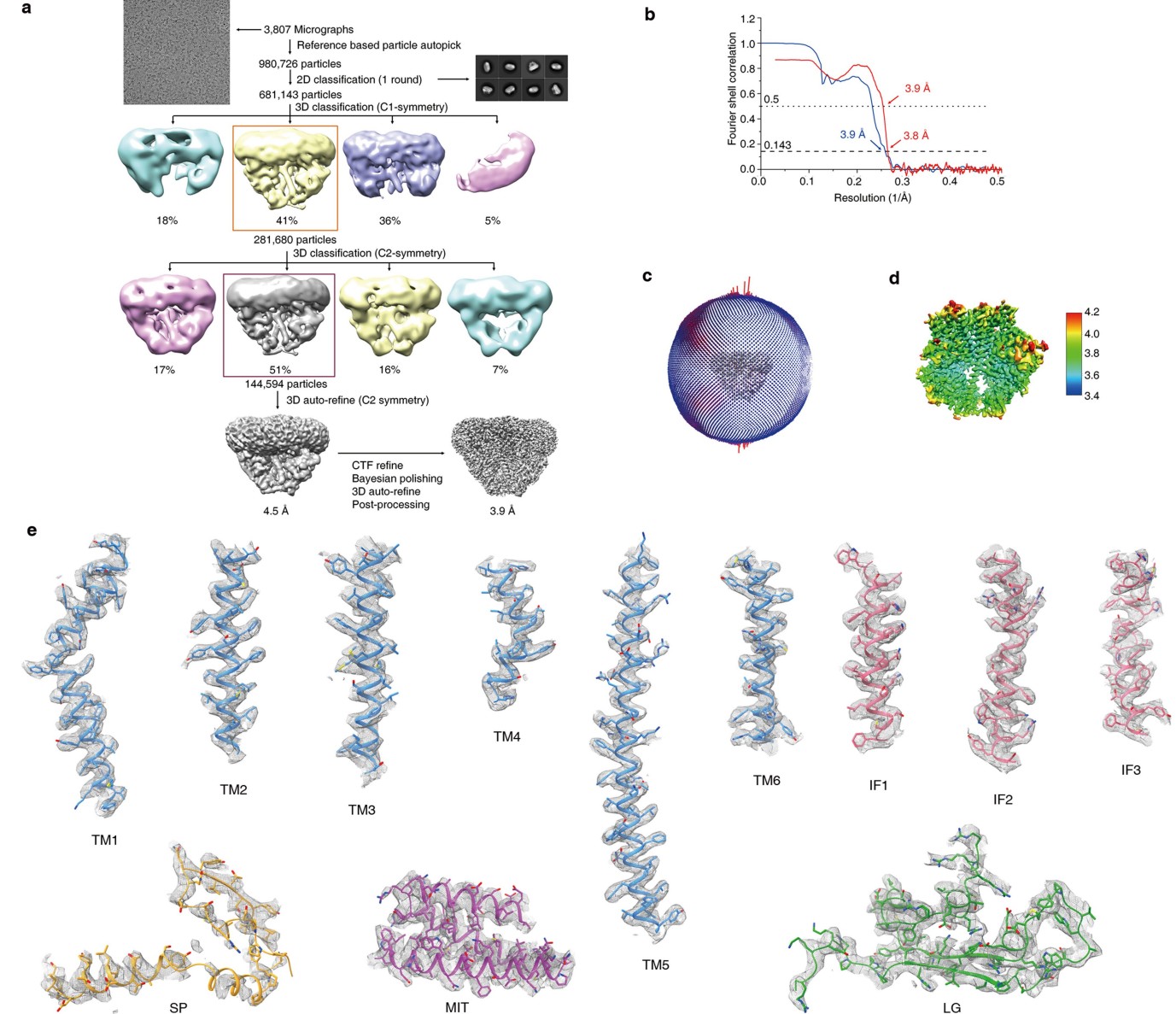

**Extended Data Fig. 5 | Cryo-EM data analysis of UDP/(GlcNAc)₃-bound**
**PsChs1. a**, Flowchart of cryo-EM data processing. **b**, FSC curves of UDP/
(GlcNAc)₃-bound PsChs1, including FSC between two independently refined
half-maps generated by RELION (blue) and model-to-map FSC generated by
Phenix (red). **c**, Angular distribution of the particles contributing to the final
reconstruction of PsChs1. **d**, The map of UDP/(GlcNAc)₃-bound PsChs1 is
colored according to estimated local resolution. **e**, Sample maps of TM helices,
N-terminal subdomains, and IF helices of UDP/(GlcNAc)₃-bound PsChs1.

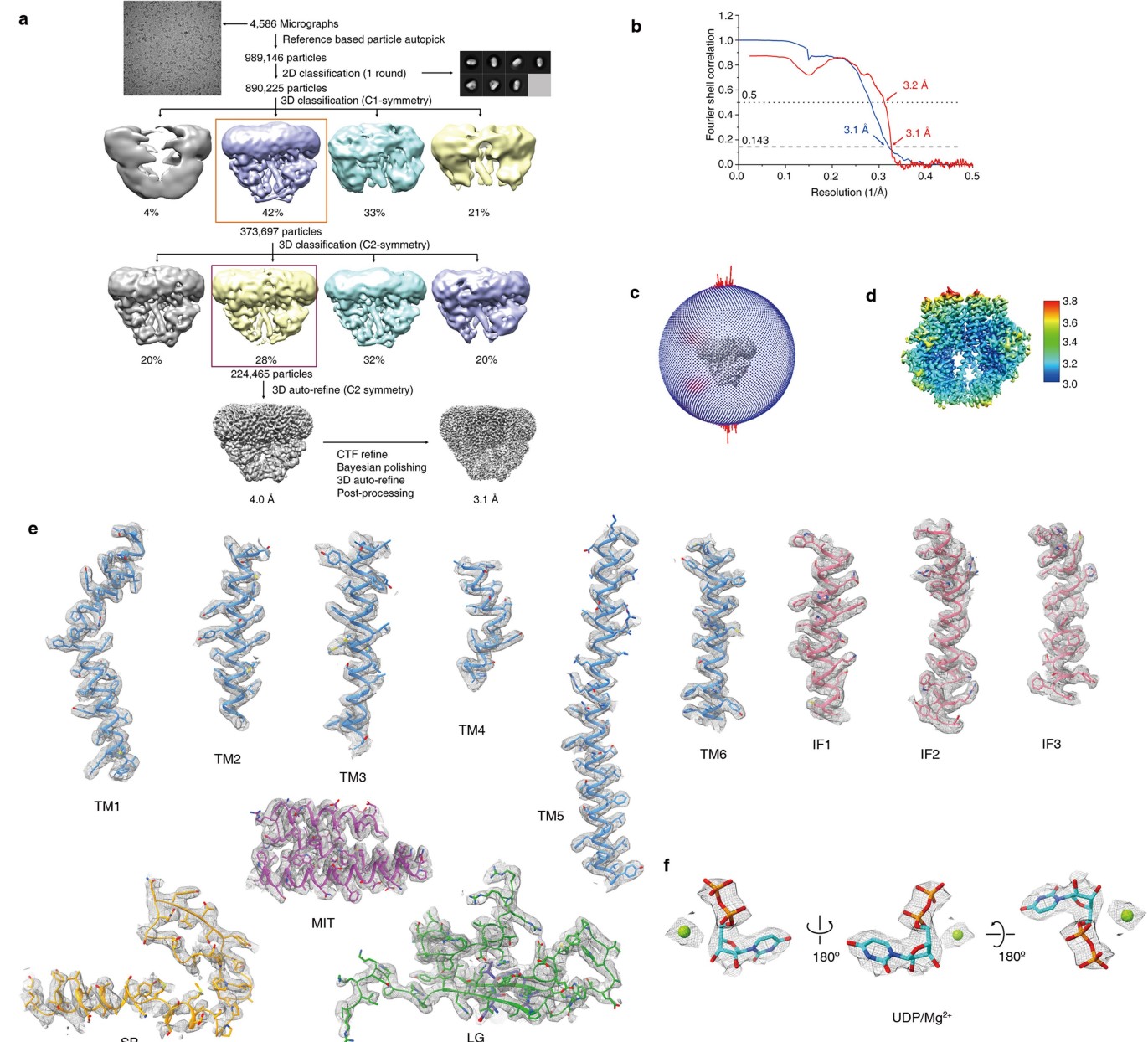

**Extended Data Fig. 6 | Cryo-EM data analysis of UDP-bound *Ps*Chs1.**
**a**, Flowchart of cryo-EM data processing. **b**, FSC curves of UDP-bound *Ps*Chs1, including FSC between two independently refined half-maps generated by RELION (blue) and model-to-map FSC generated by Phenix (red). **c**, Angular distribution of the particles contributing to the final reconstruction of *Ps*Chs1. **d**, The map of UDP-bound *Ps*Chs1 is colored according to estimated local resolution. **e**, Sample maps of TM helices, N-terminal subdomains, and IF helices of UDP-bound *Ps*Chs1. **f**, EM density of UDP/Mg²⁺.

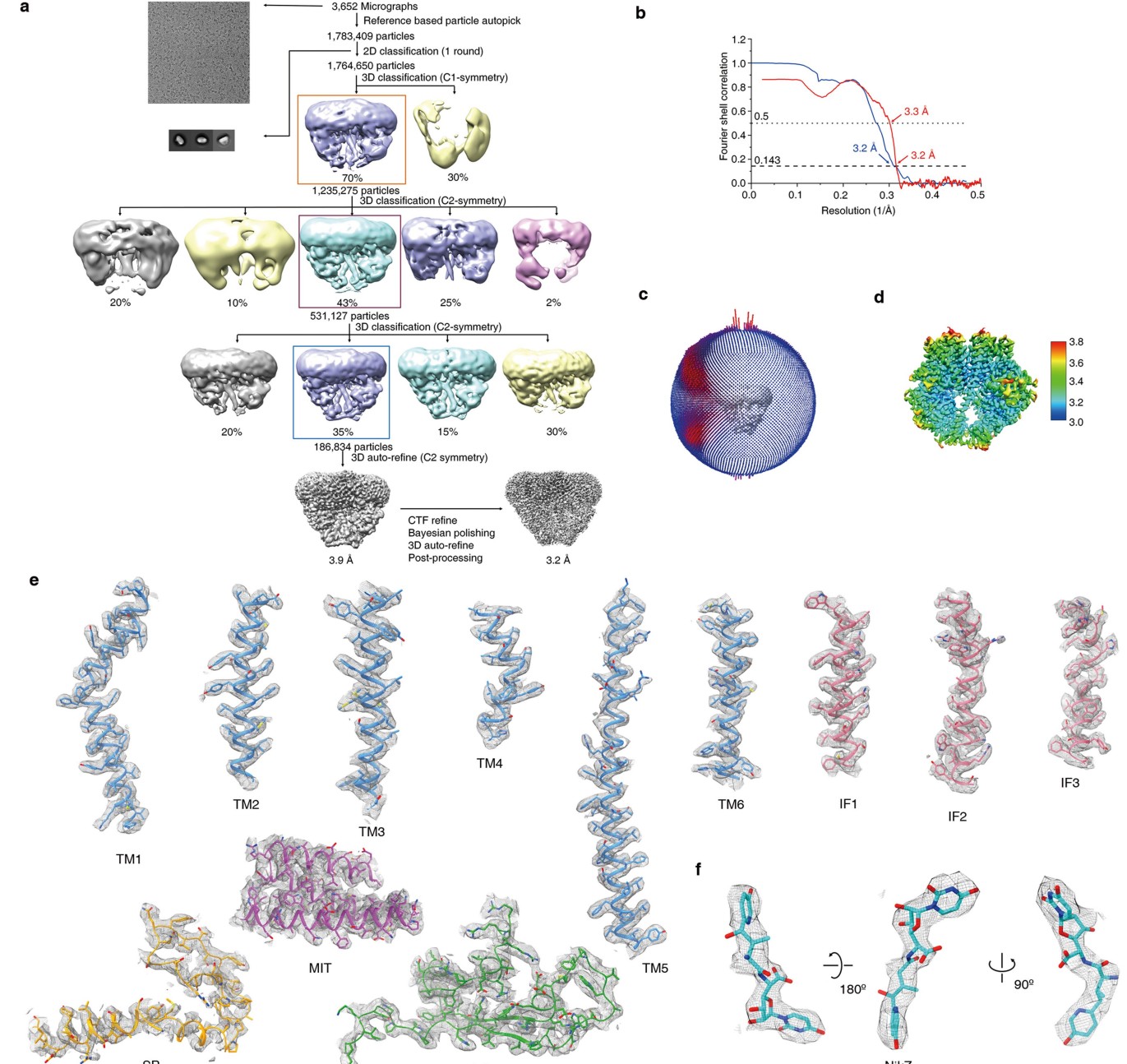

**Extended Data Fig. 7 | Cryo-EM data analysis of NikZ-bound *Ps*Chs1.**
**a**, Flowchart of cryo-EM data processing. **b**, FSC curves of NikZ-bound *Ps*Chs1, including FSC between two independently refined half-maps generated by RELION (blue) and model-to-map FSC generated by Phenix (red). **c**, Angular distribution of the particles contributing to the final reconstruction of *Ps*Chs1. **d**, The map of NikZ-bound *Ps*Chs1 is colored according to estimated local resolution. **e**, Sample maps of TM helices, N-terminal subdomains, and IF helices of NikZ-bound *Ps*Chs1. **f**, EM density of NikZ.

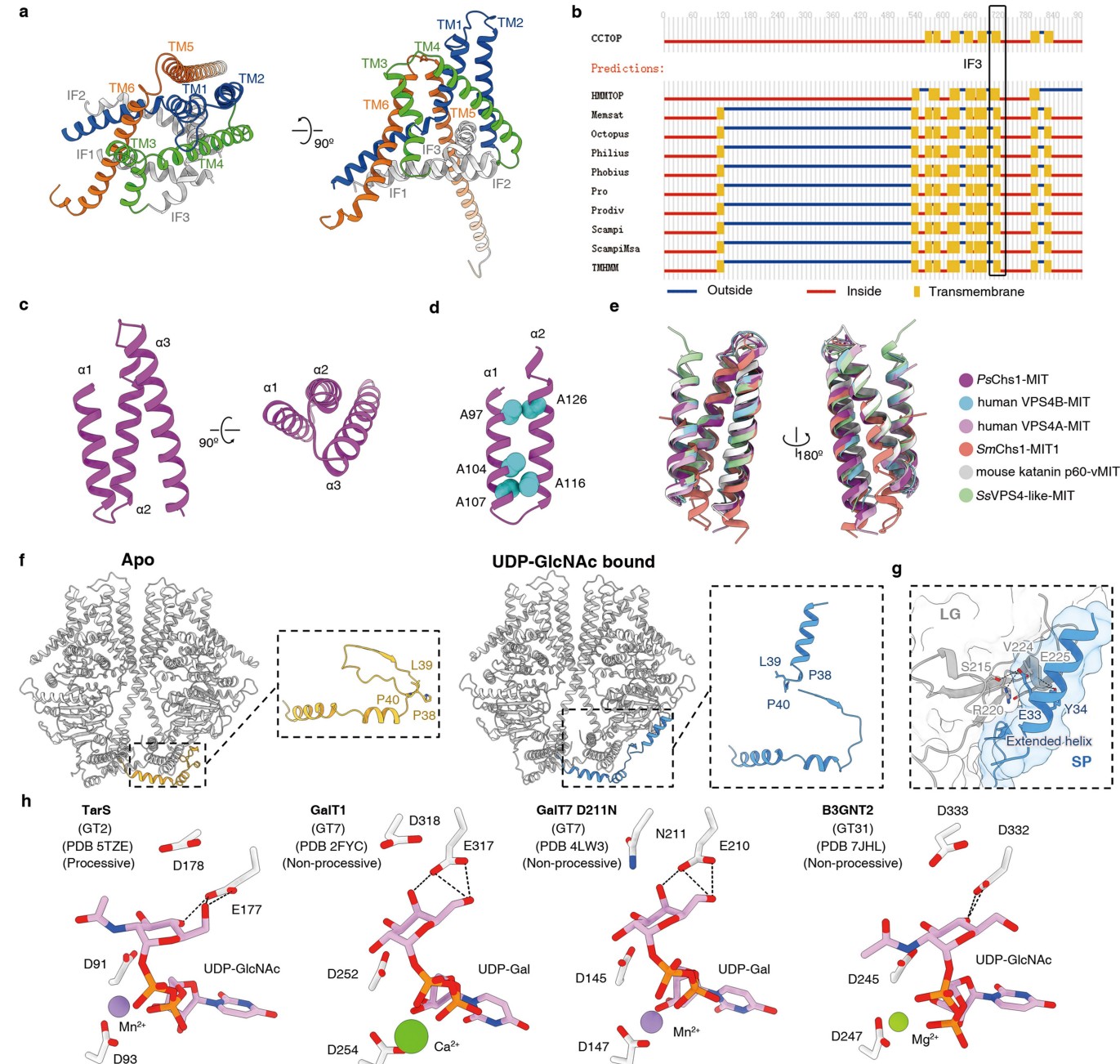

**Extended Data Fig. 8 | Architectures of *Ps*Chs1. a**, The assemblies of six TM helices (TM1–6) and three IF helices (IF1–3). Note that IF3 is indeed not a membrane spanning helix but forms a bent helix parallel to the cytosolic side of the membrane. **b**, TM topology of *Ps*Chs1 according to various topology and topography prediction methods. IF3 is frequently predicted to be a TM helix. **c**, Ribbon diagram of the *Ps*Chs1 MIT subdomain (residues 94–162). **d**, The "alanine zipper" connecting *Ps*Chs1 MIT helices α1 and α2. The five alanine side chains within this motif are specifically shown. **e**, Structural alignments of the *Ps*Chs1 MIT subdomain with other known MIT domains downloaded from the PDB database. The compared structures included the following: human

VPS4B-MIT (1WR0), human VPS4A-MIT (1YXR), *Saprolegnia monoica* CHS1 MIT1 (*Sm*Chs1-MIT1; 2MPK), mouse katanin p60-vMIT (2RPA), and *Saccharolobus solfataricus* VPS4-like-MIT (2V6Y). **f**, Structural comparison between apo *Ps*Chs1 and UDP-GlcNAc-bound *Ps*Chs1 showed that the extended N-terminal region in UDP-GlcNAc-bound *Ps*Chs1 represents a different conformation. **g**, The interaction interface formed between the extra helix of one protomer (blue) and the neighboring protomer (gray) is depicted. **h**, Active sites of several metal-dependent inverting GT-A fold glycosyltransferases. The hydrogen bonds between the D/E in the (D/E)DX motif and donor sugar are shown.

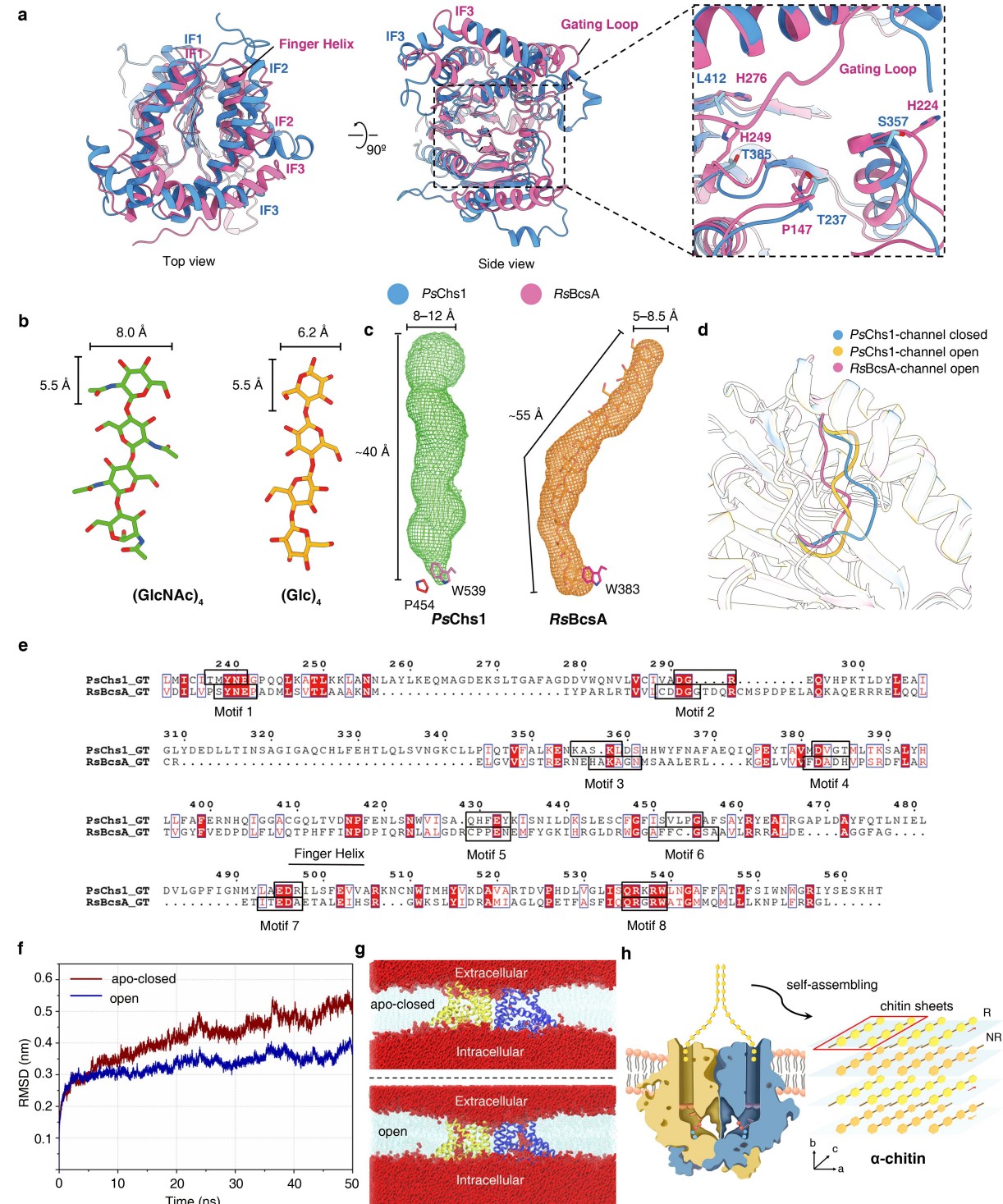

**Extended Data Fig. 9 | Comparison of *Ps*Chs1 and *Rhodobacter sphaeroides* bacterial cellulose synthase A (*Rs*BcsA) and molecular dynamics simulation. a**, The structures of *Ps*Chs1 and *Rs*BcsA are superimposed by ChimeraX and depicted in two perspectives. The catalytic domains are magnified (right). Functionally important residues are labeled and shown as sticks. **b**, Dimensions of (GlcNAc)₄ and (Glc)₄. The structures of the (GlcNAc)₄ and (Glc)₄ oligosaccharides were downloaded from the PDB database (5Y2C and 4HG6). **c**, Length and diameter of the presumed chitin (left, green mesh) and cellulose (right, orange mesh) channels of *Ps*Chs1 and *Rs*BcsA, calculated by PyMOL Caver plugin. The calculated channel of *Rs*BcsA fits perfectly with the dimension of the cellulose chain included in the *Rs*BcsA crystal structure.

**d**, Structural comparison of the VLPGA loop with BcsA's FFCGSA loop. **e**, Structure-weighted sequence alignment of the catalytic domains of *Rs*BcsA and *Ps*Chs1. Conserved motifs are labeled and indicated by black boxes. **f**, Backbone RMSD of apo *Ps*Chs1 (closed conformation) and product-bound *Ps*Chs1 (open conformation) during the MD simulation of 50 ns. **g**, From a snapshot of 50 ns, apo *Ps*Chs1 shows little permeability for water (red ball), while product-bound *Ps*Chs1 (with the removal of UDP/(GlcNAc)₃ in the simulations) shows water flow through the channel. **h**, The chitin sheets are self-assembled in an anti-parallel manner along the b-axis to finally form α-chitin.

## Extended Data Table 1 | Cryo-EM data collection, refinement and validation statistics

| | Apo-*Ps*Chs1 (EMD-32545) (PDB 7WJM ) | UDP-GlcNAc-bound *Ps*Chs1 (EMD-32546) (PDB 7WJN) | UDP/(GlcNAc)$_3$-bound *Ps*CHs1 (EMD-32917) (PDB 7X05) | UDP-bound *Ps*Chs1 (EMD-32918) (PDB 7X06) | NikZ-bound *Ps*Chs1 (EMD-32547) (PDB 7WJO) |
|---|---|---|---|---|---|
| **Data collection and processing** | | | | | |
| Magnification | 22,500 | 22,500 | 22,500 | 130,000 | 22,500 |
| Voltage (kV) | 300 | 300 | 300 | 300 | 300 |
| Electron exposure (e⁻/Å$^2$) | 60 | 60 | 60 | 60 | 60 |
| Defocus range (μm) | 1.2-2.0 | 1.2-2.0 | 1.2-2.0 | 1.2-2.0 | 1.2-2.0 |
| Pixel size (Å) | 1.07 | 1.07 | 1.07 | 1.07 | 1.07 |
| Symmetry imposed | C2 | C2 | C2 | C2 | C2 |
| Initial particle images (no.) | 2,379,125 | 1,855,418 | 980,726 | 989,146 | 1,783,409 |
| Final particle images (no.) | 161,907 | 350,132 | 144,594 | 224,465 | 186,834 |
| Map resolution (Å) | 3.3 | 3.3 | 3.9 | 3.1 | 3.2 |
| FSC threshold | 0.143 | 0.143 | 0.143 | 0.143 | 0.143 |
| Map resolution range (Å) | 3.2-7.0 | 3.1-8.8 | 3.7-10.2 | 2.9-7.4 | 3.1-7.7 |
| **Refinement** | | | | | |
| Initial mode used (PDB code) | 4HG6 | apo | apo | apo | apo |
| Model resolution (Å) | 3.3 | 3.3 | 3.9 | 3.1 | 3.2 |
| FSC threshold | 0.143 | 0.143 | 0.143 | 0.143 | 0.143 |
| Map resolution range (Å) | - | - | - | - | - |
| Map sharpening *B* factor (Å$^2$) | -107.3 | -96.6 | -145.4 | -72.7 | -96.5 |
| Model composition | | | | | |
| Non-hydrogen atoms | 12,970 | 13,054 | 13,000 | 12,964 | 12,932 |
| Protein residues | 1,614 | 1,616 | 1,602 | 1,608 | 1,602 |
| Ligands | - | 4 | 10 | 4 | 2 |
| *B* factors (Å$^2$) | | | | | |
| Protein | 65.7 | 52.1 | 44.9 | 59.9 | 54.6 |
| Ligand | - | 48.4 | 39.9 | 54.5 | 43.5 |
| R.m.s. deviations | | | | | |
| Bond lengths (Å) | 0.003 | 0.003 | 0.003 | 0.003 | 0.003 |
| Bond angles (°) | 0.477 | 0.488 | 0.456 | 0.485 | 0.477 |
| **Validation** | | | | | |
| MolProbity score | 1.45 | 1.36 | 1.54 | 1.47 | 1.54 |
| Clashscore | 3.1 | 2.9 | 4.6 | 4.0 | 4.5 |
| Poor rotamers (%) | 0.29 | 0.58 | 0 | 0.43 | 0.44 |
| Ramachandran plot | | | | | |
| Favored (%) | 95.0 | 96.0 | 95.6 | 95.9 | 95.5 |
| Allowed (%) | 5.0 | 4.0 | 4.4 | 4.1 | 4.5 |
| Disallowed (%) | 0 | 0 | 0 | 0 | 0 |

Table containing details of cryo-EM data collection, processing, and refinement including relevant statistics for all maps and models generated in this study.

# Reporting Summary

## Statistics

For all statistical analyses, confirm that the following items are present in the figure legend, table legend, main text, or Methods section.

| n/a | Confirmed | |
|---|---|---|
| ☐ | ☒ | The exact sample size (*n*) for each experimental group/condition, given as a discrete number and unit of measurement |
| ☐ | ☒ | A statement on whether measurements were taken from distinct samples or whether the same sample was measured repeatedly |
| ☐ | ☒ | The statistical test(s) used AND whether they are one- or two-sided *Only common tests should be described solely by name; describe more complex techniques in the Methods section.* |
| ☒ | ☐ | A description of all covariates tested |
| ☒ | ☐ | A description of any assumptions or corrections, such as tests of normality and adjustment for multiple comparisons |
| ☒ | ☐ | A full description of the statistical parameters including central tendency (e.g. means) or other basic estimates (e.g. regression coefficient) AND variation (e.g. standard deviation) or associated estimates of uncertainty (e.g. confidence intervals) |
| ☐ | ☒ | For null hypothesis testing, the test statistic (e.g. *F*, *t*, *r*) with confidence intervals, effect sizes, degrees of freedom and *P* value noted *Give P values as exact values whenever suitable.* |
| ☒ | ☐ | For Bayesian analysis, information on the choice of priors and Markov chain Monte Carlo settings |
| ☒ | ☐ | For hierarchical and complex designs, identification of the appropriate level for tests and full reporting of outcomes |
| ☒ | ☐ | Estimates of effect sizes (e.g. Cohen's *d*, Pearson's *r*), indicating how they were calculated |

*Our web collection on statistics for biologists contains articles on many of the points above.*

## Software and code

Policy information about availability of computer code

| Data collection | SerialEM 3.6 |
|---|---|
| Data analysis | MotionCor2 1.4.0, RELION 3.08, Gctf 1.06, UCSF Chimera 1.9, UCSF ChimeraX 1.2.5, Coot 0.8.6.1, PHENIX 1.19.2, MolProbity 4, PyMOL 2.0, GraphPad Prism 7.04, ClustalX2, Omnic 8.0, Jade 6.0, GROMACS v2019.3, LINCS, SETTLE |

For manuscripts utilizing custom algorithms or software that are central to the research but not yet described in published literature, software must be made available to editors and reviewers. We strongly encourage code deposition in a community repository (e.g. GitHub). See the Nature Portfolio guidelines for submitting code & software for further information.

## Data

Policy information about availability of data

All manuscripts must include a data availability statement. This statement should provide the following information, where applicable:
- Accession codes, unique identifiers, or web links for publicly available datasets
- A description of any restrictions on data availability
- For clinical datasets or third party data, please ensure that the statement adheres to our policy

The cryo-EM density maps have been deposited in the Electron Microscopy Data Bank (EMDB, www.ebi.ac.uk/pdbe/emdb/) with the accession codes EMD-32545 (apo), EMD-32546 (UDP-GlcNAc-bound), EMD-32917 (UDP/(GlcNAc)3-bound), EMD-32918 (UDP-bound), EMD-32547 (NikZ-bound). The atomic coordinates have been deposited in the Protein Data Bank (PDB, www.rcsb.org) with the accession codes 7WJM (apo) , 7WJN (UDP-GlcNAc-bound), 7X05 (UDP/(GlcNAc)3-bound), 7X06 (UDP-bound), 7WJO (NikZ-bound). Other structural data are available from the PDB database with the accession-codes: 5Y2C, 4HG6, 1WR0, 1YXR, 2MPK, and 2V6Y.

# Field-specific reporting

Please select the one below that is the best fit for your research. If you are not sure, read the appropriate sections before making your selection.

☒ Life sciences ☐ Behavioural & social sciences ☐ Ecological, evolutionary & environmental sciences

For a reference copy of the document with all sections, see nature.com/documents/nr-reporting-summary-flat.pdf

# Life sciences study design

All studies must disclose on these points even when the disclosure is negative.

| | |
|---|---|
| Sample size | Sample preparation protocols for cryo-EM samples are described in the methods. Freezing conditions were optimized to obtain the greatest density of mono-dispersed protein particles in the thinnest possible ice. The amount of cryo-EM micrographs collected was based on the cryo-EM time allocation and previous knowledge to estimate that the size is sufficient to generate a high-resolution density.  All activity assays were performed in triplicate unless otherwise specified. |
| Data exclusions | The initial cryo-EM images are screened manually to exclude those with low contrast, thick ice or severe ice contaminations, which is a standard procedure for cryo-EM data processing. No biochemical data have been excluded. |
| Replication | Each Cryo-EM dataset comprises thousands of copies of the complex and therefore has inherent replication. Biochemical experiments were repeated independently from 2 to 10 times, and were all successfully reproduced. |
| Randomization | Allocation of Samples/Organisms/Participants into experimental groups was not performed in this study, therefore randomization is not relevant to our study. |
| Blinding | Blinding was not utilized, and not applicable in this study. Different samples in this study are the wild type and mutant enzymes of interest. Our goal was to test the effect of mutations on enzymatic activity. When we conducted the experiment, we already know the identity of the mutants regardless of their activity. |

# Behavioural & social sciences study design

All studies must disclose on these points even when the disclosure is negative.

| | |
|---|---|
| Study description | *Briefly describe the study type including whether data are quantitative, qualitative, or mixed-methods (e.g. qualitative cross-sectional, quantitative experimental, mixed-methods case study).* |
| Research sample | *State the research sample (e.g. Harvard university undergraduates, villagers in rural India) and provide relevant demographic information (e.g. age, sex) and indicate whether the sample is representative. Provide a rationale for the study sample chosen. For studies involving existing datasets, please describe the dataset and source.* |
| Sampling strategy | *Describe the sampling procedure (e.g. random, snowball, stratified, convenience). Describe the statistical methods that were used to predetermine sample size OR if no sample-size calculation was performed, describe how sample sizes were chosen and provide a rationale for why these sample sizes are sufficient. For qualitative data, please indicate whether data saturation was considered, and what criteria were used to decide that no further sampling was needed.* |
| Data collection | *Provide details about the data collection procedure, including the instruments or devices used to record the data (e.g. pen and paper, computer, eye tracker, video or audio equipment) whether anyone was present besides the participant(s) and the researcher, and whether the researcher was blind to experimental condition and/or the study hypothesis during data collection.* |
| Timing | *Indicate the start and stop dates of data collection. If there is a gap between collection periods, state the dates for each sample cohort.* |
| Data exclusions | *If no data were excluded from the analyses, state so OR if data were excluded, provide the exact number of exclusions and the rationale behind them, indicating whether exclusion criteria were pre-established.* |
| Non-participation | *State how many participants dropped out/declined participation and the reason(s) given OR provide response rate OR state that no participants dropped out/declined participation.* |
| Randomization | *If participants were not allocated into experimental groups, state so OR describe how participants were allocated to groups, and if allocation was not random, describe how covariates were controlled.* |

# Ecological, evolutionary & environmental sciences study design

All studies must disclose on these points even when the disclosure is negative.

| | |
|---|---|
| Study description | *Briefly describe the study. For quantitative data include treatment factors and interactions, design structure (e.g. factorial, nested, hierarchical), nature and number of experimental units and replicates.* |
| Research sample | *Describe the research sample (e.g. a group of tagged Passer domesticus, all Stenocereus thurberi within Organ Pipe Cactus National Monument), and provide a rationale for the sample choice. When relevant, describe the organism taxa, source, sex, age range and any manipulations. State what population the sample is meant to represent when applicable. For studies involving existing datasets, describe the data and its source.* |
| Sampling strategy | *Note the sampling procedure. Describe the statistical methods that were used to predetermine sample size OR if no sample-size calculation was performed, describe how sample sizes were chosen and provide a rationale for why these sample sizes are sufficient.* |
| Data collection | *Describe the data collection procedure, including who recorded the data and how.* |
| Timing and spatial scale | *Indicate the start and stop dates of data collection, noting the frequency and periodicity of sampling and providing a rationale for these choices. If there is a gap between collection periods, state the dates for each sample cohort. Specify the spatial scale from which the data are taken* |
| Data exclusions | *If no data were excluded from the analyses, state so OR if data were excluded, describe the exclusions and the rationale behind them, indicating whether exclusion criteria were pre-established.* |
| Reproducibility | *Describe the measures taken to verify the reproducibility of experimental findings. For each experiment, note whether any attempts to repeat the experiment failed OR state that all attempts to repeat the experiment were successful.* |
| Randomization | *Describe how samples/organisms/participants were allocated into groups. If allocation was not random, describe how covariates were controlled. If this is not relevant to your study, explain why.* |
| Blinding | *Describe the extent of blinding used during data acquisition and analysis. If blinding was not possible, describe why OR explain why blinding was not relevant to your study.* |

Did the study involve field work? ☐ Yes ☐ No

## Field work, collection and transport

| | |
|---|---|
| Field conditions | *Describe the study conditions for field work, providing relevant parameters (e.g. temperature, rainfall).* |
| Location | *State the location of the sampling or experiment, providing relevant parameters (e.g. latitude and longitude, elevation, water depth).* |
| Access & import/export | *Describe the efforts you have made to access habitats and to collect and import/export your samples in a responsible manner and in compliance with local, national and international laws, noting any permits that were obtained (give the name of the issuing authority, the date of issue, and any identifying information).* |
| Disturbance | *Describe any disturbance caused by the study and how it was minimized.* |

# Reporting for specific materials, systems and methods

We require information from authors about some types of materials, experimental systems and methods used in many studies. Here, indicate whether each material, system or method listed is relevant to your study. If you are not sure if a list item applies to your research, read the appropriate section before selecting a response.

## Materials & experimental systems

| n/a | Involved in the study |
|---|---|
| ☐ | ☒ Antibodies |
| ☐ | ☒ Eukaryotic cell lines |
| ☒ | ☐ Palaeontology and archaeology |
| ☒ | ☐ Animals and other organisms |
| ☒ | ☐ Human research participants |
| ☒ | ☐ Clinical data |
| ☒ | ☐ Dual use research of concern |

## Methods

| n/a | Involved in the study |
|---|---|
| ☒ | ☐ ChIP-seq |
| ☒ | ☐ Flow cytometry |
| ☒ | ☐ MRI-based neuroimaging |

# Antibodies

| | |
|---|---|
| Antibodies used | Anti-FLAG M2 antibody (Sigma, Cat#A8592, RRID:AB_439702) |
| Validation | All antibodies were commercially obtained and validation reports are available on the supplier website (https://www.sigmaaldrich.cn/CN/en/product/sigma/a8592). |

# Eukaryotic cell lines

Policy information about cell lines

| | |
|---|---|
| Cell line source(s) | HEK293F (Invitrogen), validation reports are available on the supplier website (https://www.thermofisher.cn/order/catalog/product/R79007) |
| Authentication | No further authentication was performed for commercially available cell lines. |
| Mycoplasma contamination | Not tested for mycoplasma contamination. |
| Commonly misidentified lines (See ICLAC register) | No commonly misidentified cell lines were used. |

