## [Peer Review File · Nature]

Manuscript Title: Structural basis for directional chitin biosynthesis

Reviewer Comments & Author Rebuttals

Reviewer Reports on the Initial Version:

Referee expertise:

Referee #1: cryo-EM, oligosaccharides

Referee #2: biosynthesis of chitin and related oligosaccharides

Referees' comments:

Referee #1 (Remarks to the Author):

The manuscript by Wei Chen et al. reports cryo EM structures of chitin synthase from an oomycete. Chitin is one of the most abundant biopolymers on earth and performs essential functions in my arthropods, fungi, and other microorganisms. While chitin synthases (CHS) come in many forms, the catalytic mechanism is most likely evolutionarily conserved.

This study presents several CHS structures in substrate-bound, product-bound, apo, and inhibited conformations. The results are novel and certainly of interest to a broad readership. I have a few comments and suggestions that may improve the manuscript.

Major points:

- Fig. 1, chitin biosynthesis: Panels 'c' and 'd' seem inconsistent. While panel c shows chitin aggregates with distinct fibrillar features, the calcofluor stained image in panel d shows 'roundish' soft materials. Perhaps this is due to different magnifications, but should be clarified in the text.
- Line 240 and generation of a presumably monomeric CHS. The corresponding EDF 2d does not support a monomeric state of the deltaSP mutant. Hence, this claim should be removed from line 240.
- The conformational changes of the VLGP motif are very interesting, but not described well in the text. First, the authors don't stress that their apo CHS structure is the first example of a membrane-integrated processive glycosyltransferase with an apparently continuous but empty transmembrane channel. This raises the question as to how the membrane's permeability barrier is maintained in this resting state (in vivo). Based on the description of the VLGP motif, it seems that this loop closes the channel in the absence of a nascent chitin polymer. This should be discussed in more detail and also shown as a separate figure. Ideally, one could run MD simulations to analyze water flux through the channel in different conformations.
- Related to the above, the loop may not function as a ratchet as it will most likely not be able to change its conformation during the translocation of a high molecular polysaccharide.
- Fig. 4a: The map quality for the GlcNAc moiety of the bound substrate seems rather poor. This

should be shown from different angles so that the sugar density can be evaluated. Could it be that the density represents UDP molecules bound in different conformations? As modeled, what is the distance between the donor's C1 carbon and the acceptor hydroxyl, based on the (GlcNAc)₃ bound structure?

- Fig. 4c: For the product-bound state, please show the UDP and (GlcNAc)₃ density in one panel, with a continuous map over all ligands. One would expect a clear distinction between the UDP and (GlcNAc)₃ densities.

- Fig. 4c and d: Why is UDP complexed with Mn²⁺ in panel c and Mg²⁺ in panel d?

- The authors state that Asp496 most likely functions as the base catalyst during catalysis, yet use a Glu495 mutant to trap the substrate-bound state. A) Could it be that this residue is important for donor sugar recognition (some examples may exist from non-processive enzymes), and B), what is the conformation/orientation of Asp496 in the (GlcNAc)₃ bound state? Is it in close proximity to the acceptor hydroxyl? In this case, this would be the oxygen of the first glycosidic linkage, counting from the active site.

- Product-bound state: The text suggests that the product-bound state was obtained from an in vitro chitin biosynthesis reaction. If so, why is the polymer only 3 sugar units long? Are any polymer differences observed after data processing in C1?

-

Minor points:

- Fig. 2a: To highlight the functions of the individual CHS domains, it would be helpful to color the domains in panel a as shown in panel c.

- EDF9d, comparison of the VLGP loop with BcsA's FFCG loop. Please show a structural comparison alongside the sequence alignment.

- Discussion: As mentioned above, a ratcheting function of the VLGP loop seems unlikely. Therefore, could the authors discuss alternative translocation mechanisms, perhaps relative to what has been suggested for cellulose synthase?

- EDF4 and relevant sections in the main text: Based on the maps shown in EDF4 and 5, it is unclear whether the N-terminal extension of the SP domain in the UDP-GlcNAc bound state indeed represents a different conformation of this region or is 'just' better resolved in this construct. Where would this extended N-terminal region be located in the dimer?

Congratulations!

Referee #2 (Remarks to the Author):

The manuscript by Chen et al reports the cryo-EM structure of chitin synthase 1 from the oomycete *Phytophthora sojae* (PsChs1). The work shows that the protein tends to form dimers but that enzyme dimerization is not required for catalytic activity as mutations that prevent dimerization are not accompanied by a loss of glycosyl-transfer activity. Instead dimerization of PsChs1 presumably assists the formation of dimers of chitin chains as these are individually extruded from each protein monomer through channels formed by the transmembrane regions of PsChs1. Structures are also presented with different ligands bound to the active site, namely N-Acetylglucosamine (GlcNAc –

monomer of chitin), chitin oligomers, the second product of the reaction UDP, and the chitin synthase inhibitor nikkomycin Z. A model is presented that describes the mechanism of chitin formation, elongation and extrusion across the plasma membrane.

While the experimental determination of a full-length chitin synthase structure is novel, some aspects related to the mechanism of biosynthesis are not. The availability of the structure and mutants could have been exploited to address key fundamental questions that remain unresolved. Instead, some of these aspects are discussed in a circumstantial, or at times, speculative manner.

Key examples follow:

- Nikkomycin Z is a well-known structural analogue of the chitin synthase substrate UDP-GlcNAc, hence it has been described many times in the literature as a competitive inhibitor. The work on nikkomycin Z presented in this manuscript merely confirms this well-known fact and it is not obvious how these data could be exploited for the design of novel, more potent and/or specific inhibitors as stated in the manuscript.

- It is unclear whether the authors claim that GlcNAc is the actual acceptor and initiator of the reaction. There is activity in the absence of GlcNAc, which infers that GlcNAc is not required for biosynthesis and that the first molecule that binds to the active site in the absence of free GlcNAc is UDP-GlcNAc. The authors infer that in the absence of free GlcNAc, a molecule of UDP-GlcNAc is first hydrolyzed to form GlcNAc as an acceptor for further elongation. But there is no evidence that this is how synthesis starts. Instead two UDP-GlcNAc molecules could bind simultaneously to the active site to initiate the reaction. This fundamental question could have been solved by determining whether two molecules of UDP-GlcNAc can bind simultaneously in the active site.

What is the actual mode of action of free GlcNAc? It could simply be a positive effector without necessarily being involved as an acceptor. This possibility could be tested using enzyme kinetics.

On the same matter, the authors could have tested whether chito-oligosaccharides can be used as acceptors by the enzyme. This has been debated in the field and, to my knowledge, there are no data that demonstrate whether this is the case or not. The authors provide a structure of the enzyme with a chito-triose engaged in the active site. Why has this not been used to determine whether the oligosaccharide can be used as an acceptor to initiate the reaction and compare the data with free GlcNAc?

- The structures presented do not convincingly show that the cations required for activity (manganese or magnesium) do actually form a bridge between (or coordinate) the phosphate groups of the sugar donor, but the authors claim and discuss this as it was the case. Shouldn't the interaction be stronger, i.e. the cations be closer to the charge phosphate groups? Also what is the structural basis for manganese being a preferred cation to magnesium?

- The fact that the enzyme forms a dimer suggests that two monosaccharides could be added simultaneously on the same chain but this possibility has not been considered in the proposed

model. Earlier theoretical models have suggested this could be the case in different enzymes that catalyse similar reactions like, for example cellulose synthase. This would explain the stereochemistry of the 1-4-linkages, which impose a rotation by 180 degrees between adjacent residues. If this cannot be demonstrated experimentally, at least a comparison with the cellulose synthase for which a structure is available should be made to exclude or not this possible mechanism.

- In oomycete species that produce chitin, chitin is in the form of an alpha allomorph. The model suggested and discussed by the authors implies that the two chitin chains that are presumably co-polymerized and self-assemble would have a parallel orientation typical of beta-chitin. This is contradictory with data in the literature that demonstrate that chitin formed in vivo is of the alpha type. There is no discussion addressing this contradiction.

- The results in Figures 1c and 1d are not consistent. Indeed, in figure 1c the authors show a precipitate formed in vitro, which they claim is chitin. The precipitate seems very abundant after 60 minutes reaction, however the Calcofluor White detection in figure 1d suggests that there is hardly any chitin synthesized after 2 hours. What is the reason for this discrepancy? It is possible that the dye was entrapped in the mass of product formed in figure 1d rather than showing specific labelling. Calcofluor White is not very specific and the use of wheat germ agglutinin coupled to a fluorophore would have been best to demonstrate that the product formed in vitro is indeed chitin. Additionally, and perhaps even more convincing would be to perform electron diffraction (TEM) on this in vitro product to confirm that it indeed corresponds to chitin. In relation to the previous point, this would also clarify whether chitin is of the alpha or beta type as electron diffraction provides different diffraction patterns for the two allomorphs. Performing this experiment would nicely confirm the non-demonstrated hypothesis that the molecular directionality of the enzymes leads to beta chitin formation.

- The paper also contains some statements that are not quite correct. For example, lines 50-52, it is stated that chitin is microfibrillar. While this is the case in most instances, work in oomycetes has actually shown that chitin occurs in the forms of granular structures in the cell walls of oomycetes (Bulone et al. (1992) Exp Mycol 16: 8-21). This ref also shows that chitin occurs as alpha chitin in oomycetes.

The statement made lines 155 and following about the Ile722 residue seems contradictory as the authors indicate that the presence of this amino acid (as in the oomycete chitin synthase) is linked to sensitivity to different pesticides. However the authors indicate that the oomycete chitin synthases are expectedly not sensitive to the pesticides mentioned.

- The finding that the microtubule interacting and trafficking subdomain of oomycete chitin synthases adopts a triple helical structure (like MIT domains from other classes of proteins) is not new and the authors do not adequately acknowledge this fact. Indeed, the structure of two MIT domains from oomycete chitin synthases has been reported previously (see Brown et al. FEBS J. (2016) 283(16):3072-88.

- In all figures where catalytic activities are presented, it would be appropriate to use number of

moles incorporated into product so the reader can have a better idea of the levels of activity measured. Instead the authors use relative %, this is not rigorous as the experimental section does not describe any control. Relative activity to what? I also have concerns with the assay described as free GlcNAc is added to 96-well plates coated with wheat germ agglutinin. This would typically give high background even in the absence of activity as the lectin would bind excess GlcNAc, even after the washing steps described. On the same matter there is no description of the experimental conditions used for the enzyme kinetics presented in supplementary material.

- The discussion lines 401-415 is highly speculative and not supported by experimental evidence. Overall more fundamental questions could have been solved using the structure, with some additional experiments, as suggested above, instead of developing speculative statements or making rather firm conclusions that are not based on firm experimental evidence. The work does not really elucidate the mechanism of action of nikkomycin Z as this has been known for a long time (competitive inhibition). The authors also indicate in multiple instances that this work establishes the basis for the design of additional chitin synthase inhibitors to block the growth of pathogenic oomycetes and fungi, but they do not consider that this type of inhibitor might also inhibit other GlcNAc transfer reactions in the host, such as reactions involved in GPI anchor biosynthesis or protein glycosylation. This would have severe consequences that would make these inhibitors unsuitable for disease control. The statements made would have deserved more insightful consideration of what adverse effects the use of chitin synthase competitive inhibitors might have for the host.

- The language is not perfect throughout the manuscript and would need some editing.

Author Rebuttals to Initial Comments:

Referee #1 (Remarks to the Author):

The manuscript by Wei Chen et al. reports cryo EM structures of chitin synthase from an oomycete. Chitin is one of the most abundant biopolymers on earth and performs essential functions in many arthropods, fungi, and other microorganisms. While chitin synthases (CHS) come in many forms, the catalytic mechanism is most likely evolutionarily conserved.

This study presents several CHS structures in substrate-bound, product-bound, apo, and inhibited conformations. The results are novel and certainly of interest to a broad readership. I have a few comments and suggestions that may improve the manuscript.

Response: We thank the reviewer for his/her time in reviewing this manuscript and the positive comment on the novelty and importance of this work. We also thank the reviewer for the constructive comments and suggestions that have helped us improve this manuscript.

Major points:

- Fig. 1, chitin biosynthesis: Panels 'c' and 'd' seem inconsistent. While panel c shows chitin aggregates with distinct fibrillar features, the calcofluor stained image in panel d shows 'roundish' soft materials. Perhaps this is due to different magnifications, but should be clarified in the text.

Response: Thank you for pointing out this issue. Indeed, the inconsistency between panels 'c' and 'd' is due to different magnifications. The resolution of laser scanning confocal microscopy allows only imaging the aggregated fibrillar chitin, which then appears as "roundish" soft material (please see the images below). To avoid misunderstanding, we repeated the experiment in which the staining with "calcofluor white" was replaced by a staining with the chitin binding protein wheat germ agglutinin (WGA) coupled to the fluorophore FITC because WGA is more specific for chitin. We included this data as well as the original one and revised the corresponding description in main text (please see Extended Data Fig. 2c and Fig. 1c). "Under a confocal laser scanning microscopy (cLSM), chitin was specifically detected by wheat germ agglutinin coupled to the fluorophore FITC. It appeared as aggregated fibrillar material at high magnification in SEM, but as "roundish" soft material at lower magnification in cLSM (Extended Data Fig. 2c)."

- Line 240 and generation of a presumably monomeric CHS. The corresponding EDF 2d does not support a monomeric state of the deltaSP mutant. Hence, this claim should be removed from line 240.

Response: Thank you for this comment. This claim has been removed in the revised manuscript.

- The conformational changes of the VLPG motif are very interesting, but not described well in the text. First, the authors don't stress that their apo CHS structure is the first example of a membrane-integrated processive glycosyltransferase with an apparently continuous but empty transmembrane channel. This raises the question as to how the membrane's permeability barrier is maintained in this resting state (*in vivo*). Based on the description of the VLPG motif, it seems that this loop closes the channel in the absence of a nascent chitin polymer. This should be discussed in more detail and also shown as a separate figure. Ideally, one could run MD simulations to analyze water flux through the channel in different conformations.

Response: Thanks for this very valuable comment. In the revised manuscript, we added “the apo CHS structure is the first example of a membrane-integrated processive glycosyltransferase with an apparently continuous but empty transmembrane channel” in the main text.

As suggested, we have run MD simulations to analyze water fluxes through the channel in the closed (apo state) and open conformation (product-bound state with ligand removed) of this VLPGA motif (Extended Data Fig. 9f). The results suggested that in the apo state, no water molecules passed through the membrane and all water molecules moved around a well-defined region inside the matrix (Extended Data Fig. 9g and Supplementary Video 1). In contrast, the open conformation promoted water flow through the channel across the cell membrane (Extended Data Fig. 9g and Supplementary Video 2). Therefore, the VLPGA motif indeed acts as a permeability barrier that prevents water flux across the membrane.

Based on that, a new separate figure panel is added to the manuscript (Extended Data Fig. 9f, g) and the corresponding paragraph on the role of VLPGA is rewritten as following: “Molecular dynamic (MD) simulations of different *PsChs1* states, representing the closed and open conformations of the channel, suggested that the VLPGA motif indeed acts as a permeability barrier of membrane that prevents water flux across the membrane in the apo state (Extended Data Fig. 9f, g, Supplementary Videos 1, 2).” Besides, as suggested, we added a new figure panel (Fig. 4b, left bottom) to show that the loop controls the opening and closing of the channel.

- Related to the above, the loop may not function as a ratchet as it will most likely not be able to change its conformation during the translocation of a high molecular polysaccharide.

Response: Thank the reviewer for this comment. In the revised manuscript, we replaced “ratchet” by “gate lock”. We think that this loop does change its conformation during translocation of the polysaccharide, but in a mode more like a “gate” instead of a “ratchet.”

- Fig. 4a: The map quality for the GlcNAc moiety of the bound substrate seems rather poor. This should be shown from different angles so that the sugar density can be evaluated. Could it be that the density represents UDP molecules bound in different conformations? As modeled, what is the

distance between the donor's C1 carbon and the acceptor hydroxyl, based on the (GlcNAc)₃ bound structure?

Response: Thanks for this insightful suggestion. We have added a new figure (Extended Data Fig. 4f, in the revised manuscript) to illustrate the map quality of the GlcNAc moiety from different angles (please also see below). We can see that the GlcNAc moiety is linked to UDP. The density does not represent the product UDP, because the enzyme here is inactive with the mutation of the active residue Glu495 to Ala.

Superposition of the donor UDP-GlcNAc bound- and the (GlcNAc)₃ bound structures suggests that the distance between the donor's C1 carbon and the acceptor hydroxyl is 8.5 Å. This seems a too far distance for a catalyzed S_N2 displacement reaction to happen between the donor and the acceptor. But the (GlcNAc)₃ bound structure is in a post-reaction state (binding to UDP, the leaving group, not UDP-GlcNAc), and UDP-GlcNAc bound structure includes a reaction chamber residue E495A mutation, thus this distance may not reflect the real distance during catalysis.

- Fig. 4c: For the product-bound state, please show the UDP and (GlcNAc)₃ density in one panel, with a continuous map over all ligands. One would expect a clear distinction between the UDP and (GlcNAc)₃ densities.

Response: Thank you for this excellent suggestion! We have revised the figure to show the UDP and (GlcNAc)₃ density in one panel to make a clear distinction between the UDP and (GlcNAc)₃ densities (please also see below).

- Fig. 4c and d: Why is UDP complexed with Mn^{2+} in panel c and Mg^{2+} in panel d?

Response: Thank you for this justifiable question. Although our biochemical data demonstrated that Mn^{2+} is preferred for *PsChs1* activity, we tried different divalent metal ions and different incubation times in order to capture different conformations and binding states of the enzyme. Fortunately, we obtained a $(GlcNAc)_3$ -bound structure when the enzyme was incubated with the substrate for 10 min in the presence of Mn^{2+} (Fig. 4c). As Mg^{2+} is less effective to stimulate *PsChs1* activity than Mn^{2+} , we added Mg^{2+} in order to slow down the reaction rate to allow the enzyme to be captured in a chito-oligosaccharide-bound state in longer time. Unfortunately, we instead obtained the product-released state with only UDP in the active site (shown in Fig. 4d).

- The authors state that Asp496 most likely functions as the base catalyst during catalysis, yet use a Glu495 mutant to trap the substrate-bound state. A) Could it be that this residue is important for donor sugar recognition (some examples may exist from non-processive enzymes), and B), what is the conformation/orientation of Asp496 in the $(GlcNAc)_3$ bound state? Is it in close proximity to the acceptor hydroxyl? In this case, this would be the oxygen of the first glycosidic linkage, counting from the active site.

Response: Thanks for these insightful comments. A) Indeed, we constructed two mutants, D496A and E495A, both of which were enzymatically inactive (Extended Data Fig. 2g). The expression level of E495A was much higher than that of D496A, hence we used E495A to trap the substrate-bound state. A benefit of using E495A is that we were able to keep the Asp496, the presumed base catalyst.

The role of Glu495 is intriguing. In the apo structure, it adopts a conformation different from those in product bound and post-synthesis states, which may reflect a “ready” conformation for binding and correctly positioning of a donor substrate at the substrate binding site. Based on structures of different substrate-bound glycosyltransferases, which show strong interactions between the donor substrate and the glutamate residue (please see below), it is indeed possible that this residue could be critical for donor sugar recognition (Ramakrishnan *et al.*, *J. Mol. Biol.* 2006; Tsutsui *et al.*, *J. Biol. Chem.* 2013; Sobhanifar *et al.*, *PLoS Pathog.* 2016; Hao *et al.*, *J. Biol. Chem.* 2020).

Active sites of several metal-dependent inverting GT-A fold glycosyltransferases. The hydrogen bonds between the D/E in the (D/E)DX motif and donor sugar are shown.

B) We found that Asp496 is in a conformation with the carboxyl pointing to the reaction chamber in four of the structures we solved, except the donor substrate bound structure whose Asp496 side chain is away from the reaction chamber (Fig. 4b, e). The side chain of Asp496 in the (GlcNAc)₃ bound state is pointing to the reaction chamber, with a distance of 6.8 Å to the hydroxyl of the last sugar unit (the nearest to the reaction chamber). Because this structure is not in a pre-synthesis and donor substrate bound state (although we believe that the last sugar unit can be an acceptor, but the substrate site is bound a UDP leaving group, which indicate a post-reaction state), this distance does not represent a suitable distance for a general base residue to catalyze a S_N2 reaction, which requires a distance (hydrogen bond) about 3 Å or even shorter. The general base Asp is supposed to bind to the hydrogen of the acceptor hydroxyl to facilitate the nucleophilic attack on the donor substrate to induce the leave of the UDP moiety and the formation of the beta-1,4-glycosidic linkage. In this case, the oxygen of the acceptor hydroxyl would be the oxygen of the first glycosidic linkage, counting from the active site.

- Product-bound state: The text suggests that the product-bound state was obtained from an in vitro chitin biosynthesis reaction. If so, why is the polymer only 3 sugar units long? Are any polymer differences observed after data processing in C1?

Response: Thanks for this good question. It is interesting why an oligosaccharide of exactly three sugar units was trapped, but we don't really know. We have designed and performed various experiments in order to capture longer oligosaccharides, but failed to obtain quality structures. As all the three sugar units show interactions with the enzyme, it is not a surprise that the (GlcNAc)₃ bound structure may represent the most stable *PsChs1*-product complex under our experimental conditions.

Structures constructed by imposing C1 revealed an almost identical structure with a trisaccharide in the channel. The conformation of each sugar unit is also identical in the structure compared to that with a C2 symmetry. You may see some subtle differences in the density maps, but they do not make any difference in model building and structure interpretation (please see the figure below).

The density of the product after data processing in C1 (right) and C2 symmetry (left) respectively.

-Minor points:

- Fig. 2a: To highlight the functions of the individual CHS domains, it would be helpful to color the domains in panel a as shown in panel c.

Response: Thanks! We revised Fig. 2a to highlight the individual CHS domains

- EDF9d, comparison of the VLPG loop with BcsA's FFCG loop. Please show a structural comparison alongside the sequence alignment.

Response: Thank you for this suggestion. We added a structural comparison of the VLPGA loop with BcsA's FFCG loop in Extended Data Fig. 9d, alongside with the sequence alignment.

- Discussion: As mentioned above, a ratcheting function of the VLPG loop seems unlikely. Therefore, could the authors discuss alternative translocation mechanisms, perhaps relative to what has been suggested for cellulose synthase?

Response: We agree with the reviewer's comment. As we mentioned in an above response, open and closed conformations of this VLPGA loop in apo and product-bound states, respectively, suggested that this loop functions more like a "gate lock" that blocks water molecules (and probably other small molecules as well) from travelling across the membrane in the apo state, prevents the substrate from leaving, and directing the product polymer into the entrance of the channel for translocation. Interestingly, we observed the corresponding residues of FFCGS in the polysaccharide-bound cellulose synthase *RsBcsA* (PDB ID: 4HG6). While this FFCGS sequence shows some level of conservation with VLPGA motif of CHSs, it is noticeable that Pro in the VLPGA motif can adopt specific conformations different from Cys and other non-proline residues, which seems consistent with our mutational data that substitution of P454 with alanine abolishes the enzyme activity. It is, therefore, unlikely that FFCGS motif in BcsA could function in the same mode as the VLPGA "gate lock" loop in CHSs.

The translocation mechanism of the cellulose synthase *RsBcsA* involves a “finger helix” and a “gating loop”, which undergoes a large conformational change during the translocation of the nascent cellulose. Structurally, *PsChs1* harbors a similar α -helix (residues 496–504) and a presumed loop (residues 736–759), which correspond to the “finger helix” and “gating loop” of *RsBcsA*, respectively (Extended Data Fig. 9a). However, we observed no significant conformational changes in the α -helix induced by binding to either the substrate or the nascent chitin chain in our structures. The corresponding “gating loop” in *PsChs1* is only partially resolved, which makes it difficult to see any conformational changes. Whether or not the “finger helix” and the “gating loop” in CHSs share comparable functions as those in *RsBcsA* is still unknown and requires further investigation.

We have revised the corresponding text in the discussion section based on above analysis.

- EDF4 and relevant sections in the main text: Based on the maps shown in EDF4 and 5, it is unclear whether the N-terminal extension of the SP domain in the UDP-GlcNAc bound state indeed represents a different conformation of this region or is ‘just’ better resolved in this construct. Where would this extended N-terminal region be located in the dimer?

Response: Structural comparison between the apo and the UDP-GlcNAc bound states revealed that residues Pro38, Leu39 and Pro40 are in different positions, indicating that the N-terminal extension of the SP domain in the UDP-GlcNAc bound state adopts a different conformation (please see the figure below). The extended N-terminal helix in UDP-GlcNAc bound state is located in the peripheral regions of the dimer and stabilized by interacting with the LG subdomain of the opposite protomer (Extended Data Fig. 8g). We have added a new figure panel (Extended Data Fig. 8f) in the revised manuscript and revised the corresponding part in the main text.

Structural comparison between apo *PsChs1* and UDP-GlcNAc bound *PsChs1*

Referee #2 (Remarks to the Author):

The manuscript by Chen *et al* reports the cryo-EM structure of chitin synthase 1 from the oomycete *Phytophthora sojae* (*PsChs1*). The work shows that the protein tends to form dimers but that enzyme dimerization is not required for catalytic activity as mutations that prevent dimerization are not accompanied by a loss of glycosyl-transfer activity. Instead dimerization of *PsChs1* presumably

assists the formation of dimers of chitin chains as these are individually extruded from each protein monomer through channels formed by the transmembrane regions of *PsChs1*. Structures are also presented with different ligands bound to the active site, namely N-Acetylglucosamine (GlcNAc – monomer of chitin), chitin oligomers, the second product of the reaction UDP, and the chitin synthase inhibitor nikkomycin Z. A model is presented that describes the mechanism of chitin formation, elongation and extrusion across the plasma membrane.

While the experimental determination of a full-length chitin synthase structure is novel, some aspects related to the mechanism of biosynthesis are not. The availability of the structure and mutants could have been exploited to address key fundamental questions that remain unresolved. Instead, some of these aspects are discussed in a circumstantial, or at times, speculative manner.

Response: We thank the reviewer for the comments on the novelty and importance of this manuscript. We appreciate the reviewer's thorough reading of this manuscript and the insightful comments and constructive suggestions, which have helped us improve this manuscript. We have revised the manuscript accordingly.

Key examples follow:

- Nikkomycin Z is a well-known structural analogue of the chitin synthase substrate UDP-GlcNAc, hence it has been described many times in the literature as a competitive inhibitor. The work on nikkomycin Z presented in this manuscript merely confirms this well-known fact and it is not obvious how these data could be exploited for the design of novel, more potent and/or specific inhibitors as stated in the manuscript.

Response: Thank you for the comments. We agree with the reviewer that the competitive inhibition mechanism of the CHS by NikZ has been previously demonstrated by biochemical data, and we now confirm this inhibition mechanism by solving the *PsChs1*-NikZ complex structure. We have changed the relevant description in the main text. This 3.2 Å structure, however, provides near-atomic resolution structural details of how nikkomycin Z interacts with the enzyme and blocks not only the substrate binding site, but also a large part of the reaction chamber and the entrance of the product translocating channel of the enzyme.

We understand the reviewer's concern about the structural data being exploited for the design of novel, more potent and/or specific inhibitors. While we are not drug design scientists, this structure can indeed be used as a template for structure-based drug design (SBDD) to discover new inhibitors which can be developed as potential anti-fungi drug candidates. However, such a drug discovery process is usually a decade-long process with very low success rate.

Again, we thank the reviewer for the comments and have revised the main text accordingly.

- It is unclear whether the authors claim that GlcNAc is the actual acceptor and initiator of the reaction. There is activity in the absence of GlcNAc, which infers that GlcNAc is not required for biosynthesis and that the first molecule that binds to the active site in the absence of free GlcNAc is

UDP-GlcNAc. The authors infer that in the absence of free GlcNAc, a molecule of UDP-GlcNAc is first hydrolyzed to form GlcNAc as an acceptor for further elongation. But there is no evidence that this is how synthesis starts. Instead two UDP-GlcNAc molecules could bind simultaneously to the active site to initiate the reaction. This fundamental question could have been solved by determining whether two molecules of UDP-GlcNAc can bind simultaneously in the active site.

Response: Thanks for this thoughtful comment. We agree with the reviewer's comment and have revised the corresponding text in discussion section and clarified the role of GlcNAc as an acceptor that initiates chitin biosynthesis.

GlcNAc, as an acceptor, can be exogenously added or generated by hydrolysis of UDP-GlcNAc. We have performed an additional experiment that showed exogenously added GlcNAc does increase the activity of *PsChs1*, but any of its oligomers does not (Extended Data Fig. 2d). We also proved by enzyme kinetics that free GlcNAc is an acceptor to initiate the biosynthesis, but is not a positive effector that increases the activity. We found that the plot of *PsChs1* activity vs donor substrate concentration fits well with the Michaelis-Menten equation with a Hill coefficient of 1, indicating that GlcNAc is not a positive effector for *PsChs1* (Extended Data Fig. 2e). This “self-priming” mechanism has been proposed by Orlean *et al.* (*Cell Surf.* 2019) based on biochemical data, which showed that chitin synthase incorporates [¹⁴C]GlcNAc into (GlcNAc)₂ and into insoluble chitin chains in a UDP-GlcNAc-dependent manner. Similar mechanism has been proposed for cellulose synthase (Morgan *et al.*, *Nat. Struct. Mol. Biol.* 2014; McManus *et al.*, *ACS Omega* 2018), and was very recently suggested for a viral homolog of hyaluronan synthase, a GT2 family enzyme (Maloney *et al.*, *Nature.* 2022).

The hypothesis that two UDP-GlcNAc molecules could bind simultaneously to the active site to initiate the reaction implies that the reaction between two UDP-GlcNAc molecules could generate a UDP-linked disaccharide as an initiator. However, the formation of a UDP-linked disaccharide requires a reducing end extension mechanism, which is inconsistent with the proposed chain elongation mechanism via the non-reducing end for chitin synthase (Kamst *et al.*, *Biochemistry* 1999; Sugiyama *et al.*, *J. Mol. Biol.* 1999). Moreover, the spatial restriction in the substrate-binding pocket allows only one UDP-GlcNAc molecule to bind to the active site.

What is the actual mode of action of free GlcNAc? It could simply be a positive effector without necessarily being involved as an acceptor. This possibility could be tested using enzyme kinetics.

Response: Thank you for your valuable suggestion. As our response to the previous comment, our structural and biochemical data confirmed GlcNAc is an acceptor that initiate chitin biosynthesis. We have determined the enzyme kinetics in the presence and absence of free GlcNAc. The results revealed that in both cases the plot of enzyme activity vs donor substrate concentration fits well with the Michaelis-Menten equation with a Hill coefficient of 1, suggesting that GlcNAc is not a positive effector for *PsChs1* (Extended Data Fig. 2e, please also see the data below).

The Michaelis-Menten curve (left) and Hill plot (right) of *PsChsI* in the presence (▲) or absence (○) of free GlcNAc.

On the same matter, the authors could have tested whether chito-oligosaccharides can be used as acceptors by the enzyme. This has been debated in the field and, to my knowledge, there are no data that demonstrate whether this is the case or not. The authors provide a structure of the enzyme with a chito-triose engaged in the active site. Why has this not been used to determine whether the oligosaccharide can be used as an acceptor to initiate the reaction and compare the data with free GlcNAc?

Response: Thanks for this great idea. We have tested the effects of the oligomers (GlcNAc)₂₋₅ on enzyme activity. The results indicated that, while addition of free GlcNAc increases chitin synthesis, its oligomers (GlcNAc)₂₋₅ do not affect the catalyzed chitin synthesis (Extended Data Fig. 2d, please also see the data below). This data suggested that GlcNAc can either act as an acceptor to initiate the reaction thus increase chitin biosynthesis or can be a positive effector to increase the enzyme activity, but its oligomers cannot. Our enzyme kinetics data (as described in main text and included in our response to a previous comment) excluded the possibility of GlcNAc being a positive allosteric effector. Therefore, we clarified that the monomer GlcNAc acts as an acceptor to initiate the reaction.

Actually, our structures also suggested (GlcNAc)₂ or a longer oligomer is not likely to be able to enter the active center to initiate the reaction when the enzyme is in an apo or pre-synthesis state in which the translocation channel is closed by the VLPGA motif. When this motif is in the closed conformation, the space around the catalytic center is not big enough to allow a big acceptor like (GlcNAc)₂ or a longer oligomer to be positioned. Based on the proposed general base catalyzed S_N2 displacement mechanism, an acceptor needs to interact with the general base (Asp496) with its hydroxyl at the non-reducing end for reaction initiation. Based on this we docked a mono GlcNAc in this space with the hydroxyl at the non-reducing end at a distance about 3 Å from Asp496. We can see a monomer can nicely sit at the catalytic center, but there is no more space for another GlcNAc (see the following image). This space is connected to the translating channel, and an oligomer can be positioned when the VLPGA gate lock is open.

- The structures presented do not convincingly show that the cations required for activity (manganese or magnesium) do actually form a bridge between (or coordinate) the phosphate groups of the sugar donor, but the authors claim and discuss this as it was the case. Shouldn't the interaction be stronger, i.e. the cations be closer to the charge phosphate groups? Also what is the structural basis for manganese being a preferred cation to magnesium?

Response: Thank you for pointing to these issues. We have clarified in the revised manuscript that in the structure of UDP/(GlcNAc)₃ bound *PsChs1*, the manganese ion formed coordinate bonds with the beta phosphate group of the donor, with the bond length of 2.2-2.3 Å (please see the following figure), which is consistent with the range of coordinate bonds between a first transition metal and a nitrogen- or oxygen-containing compound. The Mn²⁺ ion in the donor substrate-bound complexes only formed charge interaction with the phosphate groups, with bond distance longer than 3.5 Å, which is definitely weaker than a coordinate bond. Mg²⁺ ion, different from Mn²⁺, is not able to form any coordinate bond because it does not have empty 3d orbitals, thus can only form charge interaction with phosphate groups as we observed in the UDP-bound structure. Please see the following figure panels for detailed bond lengths (added as new Fig 4 panels in the revised manuscript).

As we mentioned above, our structures showed the bond between manganese ion and the phosphate groups in UDP/(GlcNAc)₃-bound structure was much shorter than that in UDP-GlcNAc bound structure because of the different bond types. This may represent the different divalent ion binding modes in different chitin biosynthesis states (post-catalysis and pre-catalysis), and confirms that these metal ions play an important role in assisting the transfer of the sugar moiety from the donor substrate to a receptor in CHSs and other glycosyltransferases.

Again, Mn²⁺ is a transition metal with empty 3d electron orbitals, and a strong Lewis acid for coordinate bond formation with a phosphate group. Mg²⁺, however, is an alkaline earth metal which has no empty d orbitals, and a weak Lewis acid, and can only form charge interactions with phosphate group, which is weaker than coordinate bonds. The ability of a Mg²⁺ ion in assisting catalyzed chitin biosynthesis, therefore, is not as strong as that of a Mn²⁺ ion.

Previously published biochemical analysis of many GT-A glycosyltransferases showed that these enzymes require an essential divalent metal ion to facilitate departure of the nucleoside diphosphate-leaving group by electrostatically stabilizing the developing negative charge (Lairson *et al.*, *Annu. Rev. Biochem.* 2008). Other published data revealed that metal ions affect substrate binding in many

glycosyltransferases (Choi and Cabib, *Anal. Biochem.* 1994; Zhang *et al.*, *J. Biol. Chem.* 2001; Nielsen *et al.*, *J. Biol. Chem.* 2011; Genth *et al.*, *Toxins*, 2016; Esposito *et al.*, *J. Biol. Chem.* 2018).

The interaction between cations and the phosphate groups of the sugar donor.

- The fact that the enzyme forms a dimer suggests that two monosaccharides could be added simultaneously on the same chain but this possibility has not been considered in the proposed model. Earlier theoretical models have suggested this could be the case in different enzymes that catalyse similar reactions like, for example cellulose synthase. This would explain the stereochemistry of the 1-4-linkages, which impose a rotation by 180 degrees between adjacent residues. If this cannot be demonstrated experimentally, at least a comparison with the cellulose synthase for which a structure is available should be made to exclude or not this possible mechanism.

Response: Thank you for the comments. Earlier mechanistic models assumed that chitin and cellulose synthases possess two active sites to allow binding and conversion of sugar units with alternating orientations (Saxena *et al.*, *J. Bacteriol.* 1995). However, the sequences and structures of our enzyme as well as cellulose synthases and other related glycosyltransferases suggest only one active site for donor substrate binding due to steric restrictions (Morgan *et al.*, *Nature* 2013; Maloney *et al.*, *Nature* 2022). Another hypothesis is that two adjacent catalytic sites from different monomers assembled into an oligomeric state (Merzendorfer, *J. Comp. Physiol. B* 2006; Carpita, *Plant Physiol.* 2011). This model requires the monomers to be well arranged to form a dimer/multimer that synthesizes a single sugar chain. However, the structure of cellulose synthase trimer as well as our dimer structures revealed that the active sites from the monomers are too far away (>30 Å) from each other, and each monomer synthesizes a sugar chain independently, and the polymer product is translocated independently through its own translocating channel in a monomer.

Recent structural insights into cellulose biosynthesis explained that the 180° alternating arrangement of the sugar units can be achieved by a simple rotation of the terminal sugar unit around the glycosidic bond, eliminating the need for a dual substrate binding site (Morgan *et al.*, *Nature* 2013; Morgan *et al.*, *Nature* 2016). Similar to the cellulose synthase, *PsChs1* can only accommodate one UDP-GlcNAc due to steric restraints. Moreover, the two active sites of each monomer are located too far from each other to allow addition of two monosaccharides simultaneously on the same chain.

- In oomycete species that produce chitin, chitin is in the form of an alpha allomorph. The model suggested and discussed by the authors implies that the two chitin chains that are presumably co-

polymerized and self-assemble would have a parallel orientation typical of beta-chitin. This is contradictory with data in the literature that demonstrate that chitin formed *in vivo* is of the alpha type. There is no discussion addressing this contradiction.

Response: Thank you for your comment about the polymorphic forms of synthesized chitin. Consulting more recent literature on the structural nature of α - and β -chitin reveals an updated interpretation of the electron diffraction patterns, showing that α -chitin can adopt a parallel chain arrangement. According to high resolution synchrotron X-ray fiber diffraction data, α -chitin is arranged in sheets, which are composed of parallelly aligned chitin chains that are stabilized through their hydrophobic surface of the pyranose rings (Figure 4a in Sikorski *et al.*, *Biomacromolecules* 2009). Not the single chains but these chitin sheets were suggested to be stacked in an anti-parallel manner (Figure 4b in Sikorski *et al.*, *Biomacromolecules* 2009; Figure 7a in Ogawa *et al.*, *J. Struct. Biol.* 2011; Figure 1 in Deringer, *et al.*, *Biomacromolecules* 2016). These data suggested that the difference between α -chitin and β -chitin lies in the arrangements of chitin sheets, in anti-parallel (α -chitin) or in parallel manner (β -chitin). This hypothesis is in agreement with our structural data that two monomers enzyme parallelly arranged in a dimer synthesize and translocate two chitin chains separately and parallelly.

We determined the isomorphic type of chitin synthesized by *PsChs1* by X-ray diffraction (XRD) and attenuated total reflectance Fourier transform infrared (ATR-FTIR) spectroscopy, and found that it is α -chitin. The data is included in Fig. 1d in our revised manuscript, and is also attached below. XRD and ATR-FTIR are two independent methods that are widely used for the determination of chitin isomorphs (Tsurkan *et al.*, *Carbohydr. Polym.* 2021; Ogawa *et al.*, *J. Struct. Biol.* 2010). We have also tried electron diffraction TEM, however, the synthesized product turned out to be highly sensitive to irradiation damage. Therefore, we failed to obtain an image with the diffraction rings by using electron diffraction TEM under this circumstance.

As indicated in ATR-FTIR (Fig. 1d, left panel, please also see the data below), the synthesized chitin has the same adsorption spectrum as the shrimp α -chitin, where the characteristic C=O stretching (amide I) band at 1620–1670 cm^{-1} appears as a doublet clearly distinguishable from a singlet that appears in the crystalline β -chitin (prepared from *Satsuma tubeworms*, Ogawa *et al.*, *J. Struct. Biol.* 2011). The two separate peaks of synthesized chitin at 1652 cm^{-1} and 1622 cm^{-1} were attributed to the occurrence of the intermolecular hydrogen bond -CO...NH- and the intramolecular hydrogen bond -CO...HOCH₂, respectively, while the single peak in β -chitin at about 1625 cm^{-1} was assigned to the stretching of the CO group hydrogen bonded to amide group of the neighboring intra-sheet chain (Focher, *et al.* *Carbohydr. Polym.* 1992). These spectra strongly suggested that the synthesized chitin is in α form.

The ATR-FTIR spectra are in line with our results from XRD. As shown in the XRD profiles (Fig. 1d, right panel, please also see the data below), the four sharp diffraction peaks of synthesized chitin observed at 9.3°, 12.7°, 19.3°, and 26.4°, which corresponded to 020, 021, 110, and 013 planes, respectively, are typical crystal patterns of α -chitin (Noishiki *et al.*, *Biomacromolecules* 2003; Goodrich and Winter. *Biomacromolecules* 2007; Tsurkan *et al.* *Carbohydr. Polym.* 2021) and are closely coincident with shrimp α -chitin.

FTIR spectra (left) and XRD profiles (right) of α -chitin, β -chitin, and synthesized chitin.

Based on our dimer structures and the FTIR and XRD data, we confirmed the hypothesis proposed by Sikorski *et al.* The synthesized chitin chains are aligned in parallel along the a-axis through their hydrophobic surface of the pyranose rings to form chitin sheets, which may be further self-assembled in an anti-parallel manner along the b-axis to finally form α -chitin, the more stable chitin allomorph (Zeng *et al.*, *Biomacromolecules* 2012). The dimer *PsChs1* is in agreement with the formation of chitin sheets by parallel arrangement of single chains. We have removed the discussion about the formation of β -chitin, and added the mechanistic explanation of α -chitin formation in the discussion section based on our structural, FTIR and XRD data.

- The results in Figures 1c and 1d are not consistent. Indeed, in figure 1c the authors show a precipitate formed in vitro, which they claim is chitin. The precipitate seems very abundant after 60 minutes reaction, however the Calcofluor White detection in figure 1d suggests that there is hardly any chitin synthesized after 2 hours. What is the reason for this discrepancy? It is possible that the dye was entrapped in the mass of product formed in figure 1d rather than showing specific labelling. Calcofluor White is not very specific and the use of wheat germ agglutinin coupled to a fluorophore would have been best to demonstrate that the product formed in vitro is indeed chitin. Additionally, and perhaps even more convincing would be to perform electron diffraction (TEM) on this in vitro product to confirm that it indeed corresponds to chitin. In relation to the previous point, this would also clarify whether chitin is of the alpha or beta type as electron diffraction provides different diffraction patterns for the two allomorphs. Performing this experiment would nicely confirm the non-demonstrated hypothesis that the molecular directionality of the enzymes leads to beta chitin formation.

Response: Thanks for your suggestion. In the revised manuscript, the staining with “calcofluor white” is replaced by a staining with wheat germ agglutinin (WGA) coupled the fluorophore FITC. The inconsistency between panels ‘c’ and ‘d’ is due to the different magnifications achieved with scanning electron microscopy (SEM) and confocal laser scanning microscopy (cLSM). The resolution of cLSM allows only imaging of aggregated fibrillar chitin, which appeared as “roundish” soft materials (Extended Data Fig. 2c, please also see the images below). To avoid possible misunderstandings, we moved the cLSM images to the supplemental data (Extended Data Fig. 2c). The corresponding part has been revised as follows: We observed that the synthesized chitin appeared as a fibrous material and the amount of chitin fibers increased as the reaction time progressed, under scanning electron microscopy (SEM) (Fig. 1c). Under a confocal laser scanning microscopy (cLSM), chitin was specifically detected by wheat germ agglutinin coupled to the fluorophore FITC. It appeared as

aggregated fibrillar material at high magnification in SEM, but as “roundish” soft material at lower magnification in cLSM (Extended Data Fig. 2c).

- The paper also contains some statements that are not quite correct. For example, lines 50-52, it is stated that chitin is microfibrillar. While this is the case in most instances, work in oomycetes has actually shown that chitin occurs in the forms of granular structures in the cell walls of oomycetes (Bulone et al. (1992) *Exp Mycol* 16: 8-21). This ref also shows that chitin occurs as alpha chitin in oomycetes.

Response: Thank you for this information. We have revised the corresponding statement and added this reference (Bulone et al. (1992) *Exp Mycol* 16: 8-21).

The statement made lines 155 and following about the Ile722 residue seems contradictory as the authors indicate that the presence of this amino acid (as in the oomycete chitin synthase) is linked to sensitivity to different pesticides. However, the authors indicate that the oomycete chitin synthases are expectedly not sensitive to the pesticides mentioned.

Response: We agree with the reviewer’s comment and have removed the corresponding text in the revised manuscript.

This isoleucine-associated pesticide resistance mechanism is only found for arthropods because it is only conserved in arthropod CHSs. As indicated in Extended Data Fig.1, this residue is not conserved in oomycete chitin synthase *PeChs1*, *SmChs1* and *AaChs1*, while *PsChs1* happens to have this isoleucine. So, Ile722 of *PsChs1* is not likely relevant to the pesticide resistance.

- The finding that the microtubule interacting and trafficking subdomain of oomycete chitin synthases adopts a triple helicoidal structure (like MIT domains from other classes of proteins) is not new and the authors do not adequately acknowledge this fact. Indeed, the structure of two MIT domains from oomycete chitin synthases has been reported previously (see Brown et al. *FEBS J.* (2016) 283(16):3072-88.

Response: We have noticed the previously reported structure of the MIT domain in *Saprolegnia monoica* chitin synthase, which contains two MIT domains. Brown *et al.* solved a structure of one MIT domain (MIT1) by NMR and built a homology model for the other MIT domain (MIT2). We have cited this reference in the revised manuscript and compared *PsChs1* MIT domain with other known MIT structures. As shown in Extended Data Fig. 8e, the structural architecture of MITs is highly conserved.

- In all figures where catalytic activities are presented, it would be appropriate to use number of moles

incorporated into product so the reader can have a better idea of the levels of activity measured. Instead the authors use relative %, this is not rigorous as the experimental section does not describe any control. Relative activity to what? I also have concerns with the assay described as free GlcNAc is added to 96-well plates coated with wheat germ agglutinin. This would typically give high background even in the absence of activity as the lectin would bind excess GlcNAc, even after the washing steps described. On the same matter there is no description of the experimental conditions used for the enzyme kinetics presented in supplementary material.

Response: Thank you for this comment and the suggestions. We removed the relative activity and used the number of moles incorporated into product to express catalytic activities in all figures.

To address the reviewer's concern about the high background in the enzyme activity assays, we implemented negative controls, where denatured enzyme and substrates were added, in each assay. Introducing negative controls significantly reduced background (please see the figure below). The background was subtracted during the data analysis. We also tested the background of GlcNAc at different concentrations, which showed little effect on activity measurement (please see the figure below).

We added the description of the experimental conditions for enzyme kinetics in the supplementary material of the revised manuscript as follows: “For enzyme kinetics assay, the reaction mixtures containing 1 µg *PsChs1*, 4.5 µM–10 mM UDP–GlcNAc, 50 mM Tris-HCl buffer (pH 7.5) and 150 mM NaCl were incubated in a final volume of 100 µL in the absence or presence of 1 mM GlcNAc.”

- The discussion lines 401-415 is highly speculative and not supported by experimental evidence. Overall more fundamental questions could have been solved using the structure, with some additional experiments, as suggested above, instead of developing speculative statements or making rather firm conclusions that are not based on firm experimental evidence. The work does not really elucidate the mechanism of action of nikkomycin Z as this has been known for a long time (competitive inhibition). The authors also indicate in multiple instances that this work establishes the basis for the design of additional chitin synthase inhibitors to block the growth of pathogenic oomycetes and fungi, but they do not consider that this type of inhibitor might also inhibit other GlcNAc transfer reactions in the host, such as reactions involved in GPI anchor biosynthesis or protein glycosylation. This would have severe consequences that would make these inhibitors unsuitable for disease control. The statements made would have deserved more insightful consideration of what adverse effects the use of chitin synthase competitive inhibitors might have for the host.

Response: Thank you for your very constructive comments. In the revised manuscript, we have removed the relevant statements and pointed out potential side effects of using competitive chitin synthase inhibitors for pathogen control.

We agree that the side effects of chitin synthase inhibitors should be seriously considered when they are used to control pathogenic oomycetes and fungi. Therefore, it is required to develop inhibitors with higher specificity that target specifically at the active site in pathogen's enzyme. While all glycosyltransferases show some levels of similarity, specific structural features can still be found in the enzymes from pathogenic species which are different from those in other glycosyltransferases. These specific structural differences can be employed to design drug candidate with potential specificity to target only the enzymes from the pathogenic species. We have mentioned the potential adverse effects to host organisms when using competitive chitin synthase inhibitors for pathogen control.

- The language is not perfect throughout the manuscript and would need some editing.

Response: Thank you for the comment. We have carefully checked the language throughout the whole manuscript.

Reviewer Reports on the First Revision:

Referees' comments:

Referee #1 (Remarks to the Author):

The revised manuscript by Wei Chen et al. has addressed most concerns raised by the reviewers. However, I still have several suggestions that I hope will improve the clarity of the manuscript.

Major points:

Several sections of the text are unstructured or poorly organized. For example:

Line 140-142: Prompted by my initial suggestion, the authors inserted my comment on the importance of the empty TM channel at the beginning of the structural biology section. This seems misplaced. Perhaps a better place would be together with the description of the channel properties and water flux.

Related to the point above, the authors now show that the empty closed channel prevents water flux, while the open channel does not. The flux analysis shown in EDF9g suggests that extracellular water does not enter the channel, although the gate is near the cytoplasmic water/lipid interface. How can this be explained? Also, the length of the simulation should be stated in the caption.

The nomenclature referring to the individual structures is confusing. What is the difference between the 'post synthesis structure' (line 319), the 'UDP/post-synthesis state' (line 323), and 'product bound and post-synthesis states' (line 281). Presumably all refer to the UDP-bound state containing the GlcNAc trisaccharide.

Manganese versus magnesium coordination of nucleotides. The explanation provided in the rebuttal letter should be included in the main text.

Line 322-323: Unless the authors assume that Glu495 functions as the base catalyst, it may be best not to refer to this residue as a 'catalytic residue'. Perhaps 'functionally important' residue suffices?

Fig. 4c: Please show E495, D496 and W539 in this panel to illustrate their positions relative to the trisaccharide.

Discussion, lines 482-486: This section could be moved to line 466.

Line 517: 'An acceptor substrate bound CHS...' Should it be: 'An acceptor and substrate bound CHS..'?

Lines 530 and 532: Clarify what a- and b-axes are.

Model: Figure 1 shows that PsChs1 produces alpha-chitin in vitro containing anti-parallel chitin strands. Yet, the model presented in Fig. 6 illustrates how two polysaccharides could align in a parallel fashion. Thus, the authors should speculate on how these initial parallel filaments can form

an antiparallel fibril. I believe the statement in lines 531/532 is insufficient.

Additional discussion: The structure of a fungal CHS from *Candida albicans* has just been published in NSMB. Because this enzyme also dimerizes, it would be very informative to include a brief comparison of both dimers as a supplemental discussion and/or figure.

Minor points:

Line 142 and thereafter: The authors refer to their EM maps as 'electron density' maps. Because EM is based on elastic scattering due to Coulomb forces, these maps are better referred to as (Coulomb) potential maps or just EM maps.

Lines 312-313: 'It reflects the post-glycosylation state....'. Please change to: 'It reflects the post glycosyl transfer state....'.

Referee #2 (Remarks to the Author):

The authors have made all possible efforts to address my previous comments and the criticisms from the other reviewer. I find this revised version of the manuscript is a much-improved version of the original submission and I have only minor comments as outlined below:

- The wording used in the abstract (line 29) and the introduction (lines 56-59) would need to be modified as the text reads as if chitin was widely distributed across oomycete species, which is not really the case as the presence of chitin in oomycete cell walls has been unequivocally demonstrated in a few species only. In addition, in oomycetes at least, I do not think it can be stated that "chitin is an essential structural polymer". Again, this applies to some, but not all, species and in this case, I believe the authors mean that chitin, regardless of how little is present, is vital to the microorganism as inhibition of chitin biosynthesis is lethal. I think this could be better reflected by changing the wording. The current phrase is misleading.

- The use of the word "fibrillogenesis" is unfortunate and misleading as this word is typically associated with the formation of collagen fibers in connective tissue. I would recommend the wording is changed to, e.g., "fibril formation" or a similar phrase.

- I think the text used lines 242 onwards needs to be reworded as I believe it would be unclear to a non-specialist what is meant by "parallel mode of chitin biosynthesis". The authors mean that two chitin chains are produced concurrently by the two subunits of the CHS homodimer and that these assemble in a parallel orientation, i.e. their reducing ends point in the same direction. Please make this more explicit as the current wording is unclear. The authors have now justified very well the formation of (antiparallel) alpha-chitin from pairs of parallel dimers of individual chains.

- The text still contains numerous minor errors that need to be edited – e.g. (non-exhaustive list, so a careful check throughout is recommended), "it is the major constituent" (not constituents, line 60);

“Under a confocal laser scanning microscope”... (not microscopy, line 111); etc. I cannot list all I found as there are too many.

- The “chi” abbreviation in Fig 1 should be explained in the legend.

Author Rebuttals to First Revision:

Referee #1 (Remarks to the Author):

The revised manuscript by Wei Chen et al. has addressed most concerns raised by the reviewers. However, I still have several suggestions that I hope will improve the clarity of the manuscript.

Response: We are grateful to the reviewer for the time and effort in reviewing this manuscript and providing constructive suggestions that have helped us improve this manuscript.

Major points:

Several sections of the text are unstructured or poorly organized. For example:

Line 140-142: Prompted by my initial suggestion, the authors inserted my comment on the importance of the empty TM channel at the beginning of the structural biology section. This seems misplaced. Perhaps a better place would be together with the description of the channel properties and water flux.

Response: We have moved the description on the importance of the empty TM channel to the beginning of the description of the channel properties and water flux.

Related to the point above, the authors now show that the empty closed channel prevents water flux, while the open channel does not. The flux analysis shown in EDF9g suggests that extracellular water does not enter the channel, although the gate is near the cytoplasmic water/lipid interface. How can this be explained? Also, the length of the simulation should be stated in the caption.

Response: We thank the reviewer for this comment. To investigate into the issue, we prolonged the simulation to 100 ns and observed that the water can actually enter the channel of apo *PsChs1* from the extracellular side (please see Fig. 1a below). As indicated by RMSD profiles, the structure of apo *PsChs1* reaches equilibrium after 80 ns, later than the open *PsChs1* (please see Fig. 1b below). We compared the structures of apo *PsChs1* after 0 ns, 50 ns and 100 ns MD simulations and identified relatively large conformational changes near the extracellular entrance of the channel, which is open at both 0 and 100 ns but closed at 50 ns (please see Fig. 2 below). But throughout the simulation, the gate near the cytoplasmic water/lipid interface remains closed, which prevents the water to permeate. The upper panel of EDF9g showing the extracellular water does not enter the channel, represents a snapshot at 50 ns simulation.

We have added the length of simulation (50 ns) in the caption.

Fig. 1 MD simulations of apo *PsChs1* and product-bound *PsChs1*. **a**, A snapshot of MD simulations at 100 ns; **b**, Backbone RMSD during the MD simulation of 100 ns.

Fig. 2 Conformations of the extracellular entrance of the channel at 0, 50, and 100 ns MD simulations.

The nomenclature referring to the individual structures is confusing. What is the difference between the ‘post synthesis structure’ (line 319), the ‘UDP/post-synthesis state’ (line 323), and ‘product bound and post-synthesis states’ (line 281). Presumably all refer to the UDP-bound state containing the GlcNAc trisaccharide.

Response: To remove any ambiguities, we have simplified the nomenclatures of the structures such as UDP-bound state, UDP/(GlcNAc)₃-bound state.

Manganese versus magnesium coordination of nucleotides. The explanation provided in the rebuttal letter should be included in the main text.

Response: Thank you for this suggestion. We have added the explanation of metal ion selectivity in the main text.

Line 322-323: Unless the authors assume that Glu495 functions as the base catalyst, it may be best not to refer to this residue as a ‘catalytic residue’. Perhaps ‘functionally important’ residue suffices?

Response: We have replaced the “catalytic residue” by “functionally important residue”.

Fig. 4c: Please show E495, D496 and W539 in this panel to illustrate their positions relative to the trisaccharide.

Response: Thank you for this suggestion. We have revised the figure (Fig. 3c in the revised manuscript) to show the positions of E495, D496 and W539.

Discussion, lines 482-486: This section could be moved to line 466.

Response: We have moved lines 482-486 to line 466.

Line 517: ‘An acceptor substrate bound CHS...’ Should it be: ‘An acceptor and substrate bound CHS..’?

Response: We have replaced “An acceptor substrate bound CHS” by “An acceptor and substrate bound CHS.”

Lines 530 and 532: Clarify what a- and b-axes are.

Response: The proposed unit cell of α -chitin is orthorhombic, which has three crystallographic axes. The a-axis is perpendicular to the pyranose ring, the b-axis is parallel to the pyranose ring and represents the transverse axis of the microfibrils, and c-axis corresponds to the longitudinal axis of the microfibrils. To make this clear, we have added a figure panel (Extended Data Fig. 9h) to show what a- and b-axes are.

Model: Figure 1 shows that PsChs1 produces alpha-chitin *in vitro* containing anti-parallel chitin strands. Yet, the model presented in Fig. 6 illustrates how two polysaccharides could align in a parallel fashion. Thus, the authors should speculate on how these initial parallel filaments can form an antiparallel fibril. I believe the statement in lines 531/532 is insufficient.

Response: We have added a figure panel (Extended Data Fig. 9h) to explain how these initial parallel filaments can form an antiparallel fibril.

Additional discussion: The structure of a fungal CHS from *Candida albicans* has just been published in NSMB. Because this enzyme also dimerizes, it would be very informative to include a brief comparison of both dimers as a supplemental discussion and/or figure.

Response: Thank you for the suggestion. We have added a brief comparison of both dimers in the DISCUSSION, which is as follows: “The formation of di- and oligomeric CHS complexes has been previously reported *in vitro* and *in vivo* and is in agreement with the structural data of *PsChs1* in this study and the recently published *CaChs2*. Both dimeric *PsChs1* and *CaChs2* produce α -chitin.

Interestingly, sequence alignment between *PsChs1* and *CaChs2* or other CHSs indicates that the dimeric interface residues are highly variable (Extended Data Fig. 1a) and that the N-terminal SP and MIT subdomains, which are important for *PsChs1* dimerization, are present only in oomycete CHSs. These observations suggest that the dimerization mechanism shown in our structures may be unique to CHSs from oomycetes and some fungi, and different oligomerization strategies may have evolved for CHSs from other taxa.”

Minor points:

Line 142 and thereafter: The authors refer to their EM maps as ‘electron density’ maps. Because EM is based on elastic scattering due to Coulomb forces, these maps are better referred to as (Coulomb) potential maps or just EM maps.

Response: We have corrected the ‘electron density maps’ to ‘EM maps’.

Lines 312-313: ‘It reflects the post-glycosylation state....’. Please change to: ‘It reflects the post glycosyl transfer state....’.

Response: We have corrected this mistake.

Referee #2 (Remarks to the Author):

The authors have made all possible efforts to address my previous comments and the criticisms from the other reviewer. I find this revised version of the manuscript is a much-improved version of the original submission and I have only minor comments as outlined below:

Response: We are grateful to the reviewer for taking the time to further improve this manuscript.

- The wording used in the abstract (line 29) and the introduction (lines 56-59) would need to be modified as the text reads as if chitin was widely distributed across oomycete species, which is not really the case as the presence of chitin in oomycete cell walls has been unequivocally demonstrated in a few species only. In addition, in oomycetes at least, I do not think it can be stated that “chitin is an essential structural polymer”. Again, this applies to some, but not all, species and in this case, I believe the authors mean that chitin, regardless of how little is present, is vital to the microorganism as inhibition of chitin biosynthesis is lethal. I think this could be better reflected by changing the wording. The current phrase is misleading.

Response: Thank you for the suggestion. We have modified the corresponding description in the revised manuscript.

- The use of the word “fibrillogenesis” is unfortunate and misleading as this word is typically associated with the formation of collagen fibers in connective tissue. I would recommend the wording is changed to, e.g., “fibril formation” or a similar phrase.

Response: We have changed the wording “fibrillogenesis” to ‘fibril formation’.

- I think the text used lines 242 onwards needs to be reworded as I believe it would be unclear to a non-specialist what is meant by “parallel mode of chitin biosynthesis”. The authors mean that two chitin chains are produced concurrently by the two subunits of the CHS homodimer and that these assemble in a parallel orientation, i.e. their reducing ends point in the same direction. Please make this more explicit as the current wording is unclear. The authors have now justified very well the formation of (antiparallel) alpha-chitin from pairs of parallel dimers of individual chains.

Response: Thank you for the valuable suggestion. To make it clear, we have added an explanation for “parallel mode”.

- The text still contains numerous minor errors that need to be edited – e.g. (non-exhaustive list, so a careful check throughout is recommended), “it is the major constituent” (not constituents, line 60); “Under a confocal laser scanning microscope”... (not microscopy, line 111); etc. I cannot list all I found as there are too many.

Response: Thanks for pointing out these errors. We have made a careful check throughout the whole manuscript.

- The “chi” abbreviation in Fig 1 should be explained in the legend.

Response: The “chi” represents chitinase. We have added the explanation in the legend.